# Integrated analysis of copy number variation-associated lncRNAs identifies candidates contributing to the etiologies of congenital kidney anomalies

Yibo Lu[1,3], Yiyang Zhou[1,3], Jing Guo[1,3], Ming Qi[1], Yuwan Lin[1], Xingyu Zhang[1], Ying Xiang [1,2✉], Qihua Fu [1,2✉] & Bo Wang [1,2✉]

Congenital anomalies of the kidney and urinary tract (CAKUT) are disorders resulting from defects in the development of the kidneys and their outflow tract. Copy number variations (CNVs) have been identified as important genetic variations leading to CAKUT, whereas most CAKUT-associated CNVs cannot be attributed to a specific pathogenic gene. Here we construct coexpression networks involving long noncoding RNAs (lncRNAs) within these CNVs (CNV-lncRNAs) using human kidney developmental transcriptomic data. The results show that CNV-lncRNAs encompassed in recurrent CAKUT associated CNVs have highly correlated expression with CAKUT genes in the developing kidneys. The regulatory effects of two hub CNV-lncRNAs (*HSALNG0134318* in 22q11.2 and *HSALNG0115943* in 17q12) in the module most significantly enriched in known CAKUT genes (CAKUT_sig1, $P = 1.150 \times 10^{-6}$) are validated experimentally. Our results indicate that the reduction of CNV-lncRNAs can downregulate CAKUT genes as predicted by our computational analyses. Furthermore, knockdown of *HSALNG0134318* would downregulate *HSALNG0115943* and affect kidney development related pathways. The results also indicate that the CAKUT_sig1 module has function significance involving multi-organ development. Overall, our findings suggest that CNV-lncRNAs play roles in regulating CAKUT genes, and the etiologies of CAKUT-associated CNVs should take account of effects on the noncoding genome.

[1] Pediatric Translational Medicine Institute, Shanghai Children's Medical Center, School of Medicine, Shanghai Jiao Tong University, Shanghai 200127, China. [2] Shanghai Key Laboratory of Clinical Molecular Diagnostics for Pediatrics, Shanghai 200127, China. [3]These authors contributed equally: Yibo Lu, Yiyang Zhou, Jing Guo. ✉email: 1262975038@qq.com; qfu@shsmu.edu.cn; booew@163.com

Congenital anomalies of the kidney and urinary tract (CAKUT) refer to a collection of structural renal anomalies originated from defects in embryonic kidney development. CAKUT occur with an incidence of 1 in 500 newborns, accounting for 20 ~ 30% of congenital anomalies in the prenatal period[1]. CAKUT has been identified as the leading cause of childhood end-stage renal disease, and predispose to the development of cardiovascular disease and hypertension in adulthood. The malformations associated with CAKUT consist of ureteropelvic junction obstruction, kidney agenesis, multicystic dysplastic kidneys, kidney dysplasia, kidney hypoplasia, vesicoureteral reflux, megaureter, ectopic ureter, horseshoe kidney, duplex collecting system and posterior urethral valves[2]. Kidney development is a multi-stage process that include the induction of the ureteric bud from the nephric duct, mesenchymal-epithelial transition, branching morphogenesis, and the completion of nephron patterning[1,2]. During the kidney development period, genetic defects, environmental disturbance and their interplay contribute to the etiologies of CAKUT[2]. Copy number variations (CNVs) have been suggested as a common cause of CAKUT[2,3]. Verbitsky et al. characterized the CNV landscape of CAKUT in a large cohort and discovered that six loci including 1q21, 4p16.1-p16.3, 16q11.2, 16p13.11, 17q12, and 22q11.2 accounted for 65% of patients with CNVs enriched in known genomic disorder[4].

The pathogenicity of CNVs that encompass coding sequences is commonly interpreted based on their effect on protein coding gene dosage. While this strategy may be successful for gene discovery in Mendelian disease, the pathologies of a large proportion of CNVs associated with birth defects remain undetermined. In comparison to the small proportion of coding genes, more than 70% of the human genome can generate primary transcripts. Since most CNVs also affect noncoding genome, it is necessary to consider the pathogenic mechanism involving noncoding transcripts[5]. Long noncoding RNAs (lncRNAs), defined as noncoding transcripts longer than 200 nucleotides, have been suggested as key components controlling cell fates during development[6,7]. Recent evidence have revealed the potential contribution of lncRNAs to the etiologies of CNVs associated with birth defects. Meng et al. investigated lncRNAs within 10 CNVs associated with schizophrenia risk and identified *DGCR5* as a potential regulator of genes associated with schizophrenia[8]. Alinejad-Rokny et al. identified 47 recurrent CNVs for autism spectrum disorder and discovered that constituent coding genes and lncRNAs exhibit brain-enriched patterns of expression[9]. Our team systematically analyzed the regulatory roles of lncRNAs within CNVs (CNV-lncRNAs) associated with congenital heart disease (CHD) and revealed that *HSALNG0104472* may be responsible for cardiac defects of patients with 15q11.2 deletion[10].

CAKUT is one of the leading birth defects. However, the contribution of CNV-lncRNAs to the pathogenicity of CAKUT-associated CNVs remains elusive. We hypothesize that CNV-lncRNAs might also have regulatory effects on CAKUT genes, thus their dosage effect would affect kidney development and contribute to the etiologies of CAKUT. To test our hypothesis, we summarized recurrent CAKUT-associated CNVs and retrieved the CNV-lncRNAs. Coexpression networks involving mRNAs and lncRNAs were constructed using human kidney developmental transcriptome data from LncExpDB[11]. Based on the developmental expression pattern, we identified gene modules enriched in known CAKUT genes. Since CNVs 22q11.2 and 17q12 have been suggested occur most frequently in kidney anomalies[4], we selected the hub CNV-lncRNAs *HSALNG0134318* and *HSALNG0115943* residing in the two loci for further investigation and experimental validation. Furthermore, we also discussed the comorbidity mechanisms underlying CNV-lncRNAs in kidney, heart, and brain as CAKUT cases are often accompanied by extrarenal malformations in these organs (Fig. 1, Supplementary Data 1)[3,4,12,13].

## Results

**Retrieval of CAKUT associated CNV-lncRNAs.** We summarized 19 recurrent (identified in 2 CAKUT cases or more) CAKUT-associated CNVs identified from 2824 cases, including 3 deletions, 3 duplications and 13 deletion/duplications[3,4,12,13]. 57.89% (11/19) of these CNVs have been reported as related to extrarenal malformations such as cardiovascular defects and neurodevelopmental defects (Table 1 and Supplementary Fig. 1). A total of 8997 CNV-lncRNAs encompassed by these CNVs were retrieved according to the comprehensive annotation of human lncRNAs in LncBook[14,15] (Supplementary Data 2)[16–21].

**Construction of coexpression modules with CAKUT associated CNV-lncRNAs.** Coexpression modules composed of candidate CNV-lncRNAs and 19957 mRNAs were constructed using WGCNA on the human kidney developmental transcriptomic data ($n = 40$) from LncExpDB[11]. A total of 49 coexpression modules were identified, with 66.80% (6010/8997) CNV-lncRNAs distributed among these modules. The remaining 33.20% (2987/8997) did not cluster with any module (labeled 'gray'). Among the CNV-lncRNAs clustered in the coexpression modules, 18.94% (1704/8997) were defined as hub CNV-lncRNAs (module member (MM) ≥ 0.8, $P < 0.05$) (Supplementary Data 2).

We computed the correlation coefficients between coexpression modules and sexes/developmental stages of the donors (Supplementary Data 2). The brown (r_Developmental stage = $-0.623$, $P = 1.740 \times 10^{-5}$), turquoise (labeled as CAKUT_sig1, r_Developmental stage = $-0.582$, $P = 8.180 \times 10^{-5}$) and light-green (labeled as CAKUT_sig2, r_Developmental stage = $-0.502$, $P = 9.587 \times 10^{-4}$) modules showed high negative correlation with developmental stages (Fig. 2a and Supplementary Data 2).

**Functional enrichment analyses of CAKUT associated coexpression modules.** We summarized the known CAKUT genes ($n = 172$) that have been reported as of August 2022 (Supplementary Table 1) and tested their enrichment in the coexpression modules. Out of the 49 coexpression modules, 21 contained at least one CAKUT gene, and two modules (CAKUT_sig1, $n = 78$, $P = 1.150 \times 10^{-6}$; CAKUT_sig2, $n = 5$, $P = 0.046$) showed significant enrichment of the CAKUT genes (Fig. 2b, c and Supplementary Data 2).

To reveal the functional significance of the coexpression modules, we performed functional enrichment analysis on the protein coding genes in modules that contained at least 5 CAKUT genes. The results suggested that the largest module CAKUT_sig1, which contained 5630 protein coding genes, was enriched in functions related to CAKUT such as renal system development ($P_{adj} = 0.004$), kidney development ($P_{adj} = 0.005$), and urogenital system development ($P_{adj} = 0.006$). Additionally, the CAKUT_sig1 module also showed enrichment in functions involving multiple extrarenal systems like appendage development ($P_{adj} = 0.002$), limb development ($P_{adj} = 0.002$), in utero embryonic development ($P_{adj} = 0.005$), and ventricular septum development ($P_{adj} = 0.008$) (Fig. 2d). The CAKUT_sig2 module, which was also enriched in CAKUT genes, also showed enrichment in CAKUT-related functions of nephron development ($P_{adj} = 2.862 \times 10^{-3}$), cell junction assembly ($P_{adj} = 2.862 \times 10^{-3}$), and glomerulus development ($P_{adj} = 3.048 \times 10^{-3}$) (Fig. 2d and Supplementary Data 3).

**Validation of regulatory effects for *HSALNG0134318* and *HSALNG0115943*.** Consistent with enrichment of the CAKUT-associated functions, 49.76% (206/414) of CNV-lncRNAs in the

## CNVs increase risk of CAKUT (with unknown pathogenesis)

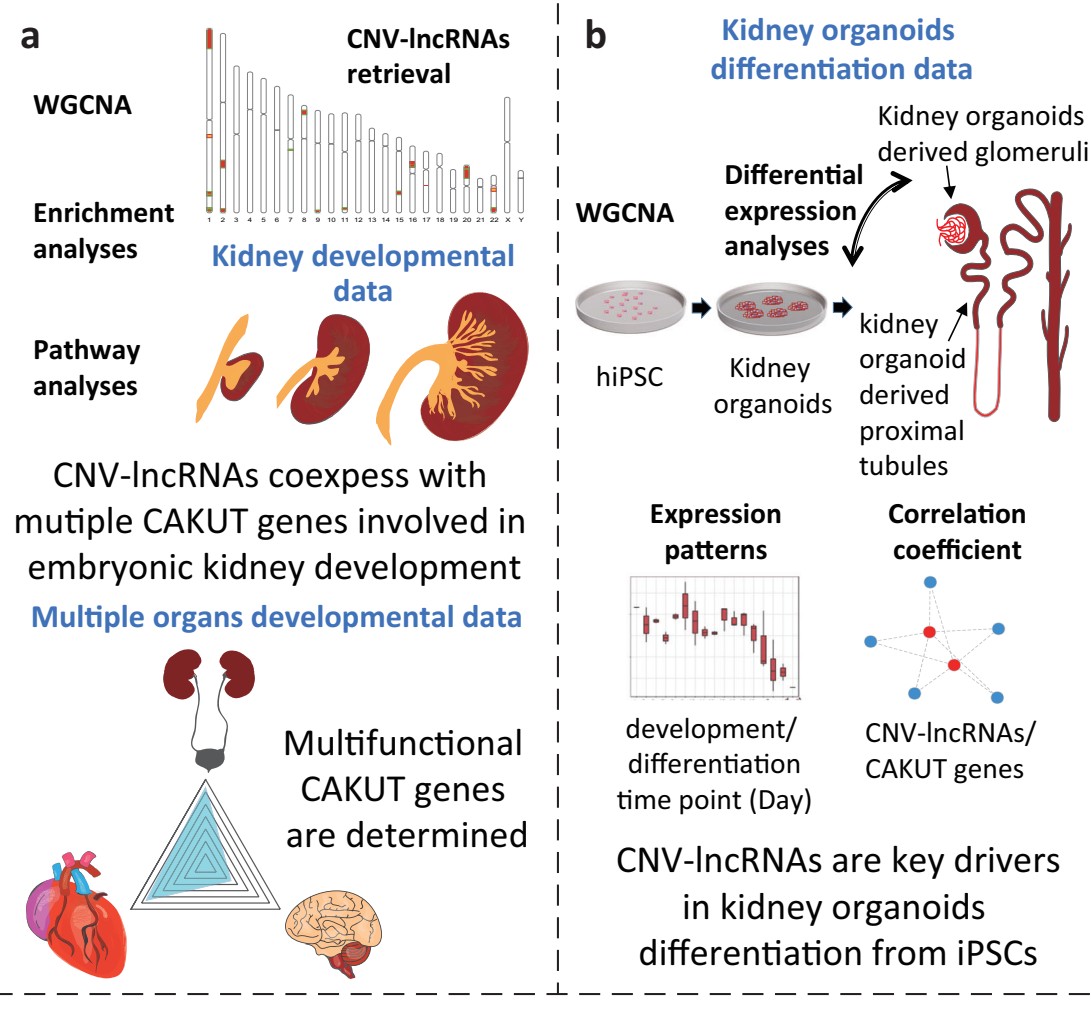

**a**

WGCNA

Enrichment analyses

Pathway analyses

CNV-lncRNAs retrieval

Kidney developmental data

CNV-lncRNAs coexpess with mutiple CAKUT genes involved in embryonic kidney development

Multiple organs developmental data

Multifunctional CAKUT genes are determined

**b**

Kidney organoids differentiation data

WGCNA

Differential expression analyses

Kidney organoids derived glomeruli

kidney organoid derived proximal tubules

hiPSC → Kidney organoids →

Expression patterns

development/ differentiation time point (Day)

Correlation coefficient

CNV-lncRNAs/ CAKUT genes

CNV-lncRNAs are key drivers in kidney organoids differentiation from iPSCs

**c** human embryonic kidney (HEK) 293 cell line

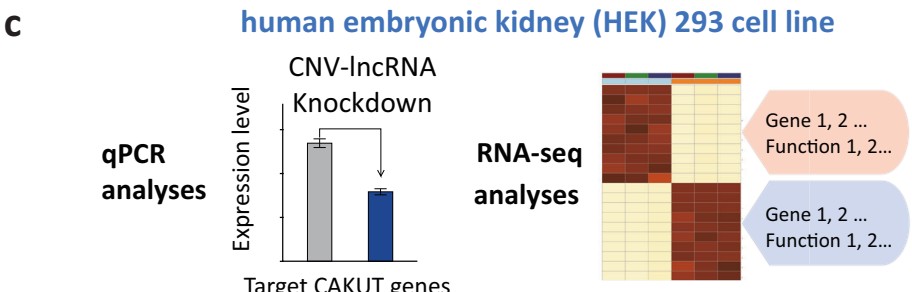

qPCR analyses

CNV-lncRNA Knockdown

Expression level

Target CAKUT genes

RNA-seq analyses

Gene 1, 2 … Function 1, 2…

Gene 1, 2 … Function 1, 2…

Deletions of CNV-lncRNAs down-regulate CAKUT gene expression in human embryonic kidney cell line

## CNV-associated lncRNAs could increase risk of CAKUT

CAKUT_sig1 module and 34.78% (8/23) of CNV-lncRNAs in the CAKUT_sig2 module showed high correlation ($|r| \geq 0.7$) with CAKUT genes (Supplementary Data 4, 5). Although CAKUT cases share developmental origins, they encompass a broad spectrum of clinical phenotypes, such as kidney anomalies (renal agenesis, hypoplasia, dysplasia and multicystic dysplasia), vesicoureteral reflux, obstructive uropathy, duplicated collecting system, posterior urethral valves, and ectopic kidney. 22q11.2 deletion and 17q12 deletion have been most linked to kidney anomalies[4]. Since we used the transcriptomic datasets of kidney development for coexpression analysis, we focused on 2 hub CNV-lncRNAs: *HSALNG0134318* located in 22q11.2 (MM = 0.87) and *HSALNG0115943* located in 17q12 (MM = 0.85) from the CAKUT_sig1 module, which were coexpressed and

**Fig. 1 Study workflow. a** 19 recurrent CNVs associated with CAKUT (pathogenic or likely pathogenic) were identified based on clinical researches of congenital anomalies of the kidney and urinary tracts (CAKUT) cases[3,4,12,13] (Table 1). We retrieved lncRNAs located within these genomic regions as candidate CNV-lncRNAs. Weighted gene coexpression network analysis (WGCNA) was performed to characterize the coexpression profile of CNV-lncRNAs and protein coding genes based on human kidney developmental transcriptomic data ($n = 40$). Downstream analyses were conducted to identify function and expression patterns of CAKUT-associated CNV-lncRNAs and their potential target CAKUT genes. **b** Based on the in vitro kidney organoids differentiation RNA-seq data ($n = 53$), the key driving roles of CAKUT-associated CNV-lncRNAs during the formation of kidney structure were verified. **c** In vitro knockdown experiments were performed to validate the predicted regulatory effects on kidney development of hub CNV-lncRNA *HSALNG0134318* and *HSALNG0115943* in human embryonic kidney cell line (HEK293), evaluated by qPCR analyses and RNA-seq analyses. hiPSCs, human induced pluripotent stem cells.

significantly correlated ($|r| \geq 0.5$, $P_{adj} < 0.05$) with multiple CAKUT genes (Fig. 3a, b and Supplementary Data 2–4). In addition, most of these coexpression relationships were also observed through expression patterns of kidney development (Fig. 3c and Supplementary Data 1, 6). Knockdown experiments of two hub CNV-lncRNAs in the human embryonic kidney (HEK293) cell line were conducted to validate their regulatory effects on the coexpressed CAKUT genes. The results showed that knockdown of these CNV-lncRNAs significantly downregulate ($P < 0.05$) CAKUT genes (Fig. 4a, b and Supplementary Data 7). Interestingly, *HSALNG0115943* was regulated by *HSALNG0134318* (Fig. 4a and Supplementary Data 7). Furthermore, RNA-seq analysis and functional enrichment analysis suggested that knockdown of *HSALNG0134318* significantly downregulate genes ($n = 839$, $P_{adj} < 0.05$) enriched in kidney development-related pathways (Fig. 4c, Supplementary Figs. 2, 3 and Supplementary Data 8).

**Driving roles of *HSALNG0134318* and *HSALNG0115943* during kidney organoids differentiation.** To further identify the roles of two hub CNV-lncRNAs in kidney development, we investigated the expression patterns of *HSALNG0134318* and *HSALNG0115943* in an RNA-seq dataset of kidney organoids differentiation from human induced pluripotent stem cells (iPSCs) (differentiation day 0–25). Noticeably, WGCNA indicated that *HSALNG0134318* (MM = 0.85) and *HSALNG0115943* (MM = 0.87) were also hub CNV-lncRNAs that clustered in modules significantly correlated ($|r| \geq 0.5$, $P_{adj} < 0.05$) with kidney organoids and early differentiation stage respectively (Fig. 5a and Supplementary Data 9). We verified the coexpression relationships between the CNV-lncRNAs and CAKUT genes previously identified in CAKUT_sig1 module: *HSALNG0134318* showed high positive correlation with *NOTCH2*, *KAT6B*, and *CDC5L* (Fig. 5b and Supplementary Data 2, 9); *HSALNG0115943* showed high positive correlation with *PSMD12*, *KDM6A*, *ANOS1*, and negative correlation with *IFT74*, *BBS4*, *IFT52*, *NRIP1*, *BBS7*, and *NIPBL* (Fig. 5c and Supplementary Data 2, 9). In another dataset consisting of 12 mature kidney organoids derived from kidney tissues (differentiation day 25–28), the correlation relationships between two CNV-lncRNAs and CAKUT genes were observed, consistent with other datasets (Supplementary Fig. 4 and Supplementary Data 10, 11).

The kidney organoids differentiation greenyellow module containing *HSALNG0134318* was significantly enriched in kidney development-related functions of renal tubule development ($P_{adj} = 0.028$), nephron tubule development ($P_{adj} = 0.028$), and distal tubule development ($P_{adj} = 0.028$) (Fig. 5d and Supplementary Data 9). Differential expression analysis of differentiated proximal tubules ($n = 3$) and glomeruli samples ($n = 6$) from the mature kidney organoids dataset suggested that *HSALNG0115943* was upregulated in the proximal tubules, whereas *HSALNG0134318* did not show a significant change (Fig. 5e and Supplementary Data 10). Differential expressed patterns of CAKUT genes were also identified (Fig. 5f and Supplementary Data 10).

**Potential synergy involving CAKUT associated CNV-lncRNAs.** To reveal the potential synergy of CAKUT associated CNV-lncRNAs with other ncRNA species, we investigated the potential contribution of competing endogenous RNA (ceRNA) mechanism driven by CNV-lncRNAs distributed in the CAKUT_sig1 and CAKUT_sig2 modules (kidney developmental WGCNA, $n = 437$). We found that CAKUT gene *AFF3* in the CAKUT_sig1 module was significantly regulated by co-expressed CNV-lncRNAs *HSALNG0148020* ($P_{adj} = 3.390 \times 10^{-4}$), *HSALNG0111125* ($P_{adj} = 4.490 \times 10^{-5}$) and *HSALNG0110138* ($P_{adj} = 1.586 \times 10^{-4}$) through the ceRNA mechanism (Supplementary Fig. 5 and Supplementary Data 12). The hub CNV-lncRNA *HSALNG0134318* was not identified as a driver in the lncRNA-miRNA-mRNA regulatory network (Supplementary Data 12). To study the potential interaction between *HSALNG0134318* and other genes within their CNV region 22q11.2, we characterized the expression of 538 lncRNA, 123 protein coding genes and 22 miRNAs in 22q11.2 (Supplementary Data 12). We found that only 5.69% (7/123) of the protein coding genes and 2.97% (16/538) of the lncRNA in the 22q11.2 region distributed in two CAKUT related modules (CAKUT_sig1 and CAKUT_sig2), of which *HSALNG0134318* was the only hub CNV-lncRNA (MM = 0.87) (Supplementary Data 12). The miRNA hsa-miR-1306-5p and hsa-miR-3198 in the 22q11.2 CNV region had potential interaction with *HSALNG0134318* (Supplementary Fig. 6 and Supplementary Data 12). Furthermore, we predicted the synergic transcription factors (TFs) of *HSALNG0134318* according to its promoter region sequence (Supplementary Table 2 and Supplementary Data 12). Among these predicted transcription factors, TFAP2A was in the CAKUT gene list (Supplementary Table 2 and Supplementary Data 12).

**Expression patterns of the CAKUT_sig1 module in kidney, heart, brain and cerebellum.** Since the CAKUT_sig1 module showed function significance involving multiple systems (Fig. 2d), we further compared the expression patterns of its CNV-lncRNAs and mRNAs in the kidney, heart, brain, and cerebellum developmental samples. Differential expressed genes in the kidney tissues were identified (Fig. 6a and Supplementary Data 13–15). Comparing with the kidney samples, genes with heart-biased expression were enriched in cardiac functions, and genes with brain/cerebellum-biased expression were enriched in neurological functions (Fig. 6b and Supplementary Data 13–15). On the other hand, the kidney-biased genes were associated with multiple organs development involving kidney development, neural development, cardiac development and other organ development (Fig. 6c and Supplementary Data 13–15). We further summarized the known congenital heart disease (CHD) genes and neurodevelopmental disorders (NDDs) genes (Supplementary Data 16). The results showed that both two gene sets were enriched in CAKUT genes (Fig. 6d and Supplementary Data 16). Among the CAKUT genes involved in these disorders, 43.8% (14/32) were distributed in the CAKUT_sig1 module (Fig. 6e, Supplementary Table 3 and Supplementary Data 2), and they showed significantly upregulated expression in the

**Table 1 Recurrent CAKUT associated CNVs.**

| CNV location | CNV type | Classification of CNV | Renal/urinary system phenotype CAKUT | Reference | Risk genes | Extrarenal malformations (Cardiovascular) | | | Extrarenal malformations (Neurocognitive) | | |
|---|---|---|---|---|---|---|---|---|---|---|---|
| | | | | | | Cardiovascular defect | Cardiovascular phenotype | Reference | Neurocognitive defect | Neurocognitive phenotype | Reference |
| **CNV landscapes across CAKUT cases** | | | | | | | | | | | |
| 10q21 | del | Intergenic CNV Likely pathogenic CNV | KA | Sanna-Cherchi, Simone et al. Am J Hum Genet. 2012[3] Westland, Rik et al. Kidney Int. 2015[13] | - | N | - | - | N | - | - |
| 12q24 | del/dup | Likely pathogenic CNV single gene CNV | KA | Sanna-Cherchi, Simone et al. Am J Hum Genet. 2012[3] Verbitsky, Miguel et al. J Clin Invest. 2015[12] | - | N | - | - | N | - | - |
| 15q11.2 | dup | likely pathogenic CNV | VUR OU PUV | Verbitsky, Miguel et al. Nat Genet. 2019[4] | - | Y | COA, TOF, PA, TAPVR | Glessner JT et al. 2014[18] | Y | SCZ, Epilepsy, ID | Marshall CR et al. Nat Genet. 2017[19] Borlot F et.al. JAMA Neurol. 2017[20] |
| 15q13 | del | Likely pathogenic CNV known CNV | KA | Sanna-Cherchi, Simone et al. Am J Hum Genet. 2012[3] Verbitsky, Miguel et al. J Clin Invest. 2015[12] Westland, Rik et al. Kidney Int. 2015[13] | - | N | - | - | Y | Epilepsy, ID, Autism Spectrum Disorder, SCZ | Borlot F et.al. JAMA Neurol. 2017[20] SFARI database[21] Marshall CR et al. Nat Genet. 2017[19] |
| 16p11 | del/dup | known CNV | KA PUV OU VUR DCS | Sanna-Cherchi, Simone et al. Am J Hum Genet. 2012[3] Verbitsky, Miguel et al. J Clin Invest. 2015[12] Verbitsky, Miguel et al. Nat Genet. 2019[4] | TBX6 | Y | TOF, BAV, PA | Pierpont, Mary Ella et al. Circulation. 2018[16] | Y | Autism Spectrum Disorder, SCZ | SFARI database[21] Marshall CR et al. Nat Genet. 2017[19] |
| 16p12 | del/dup | Likely pathogenic CNV known CNV | KA UPJO | Sanna-Cherchi, Simone et al. Am J Hum Genet. 2012[3] Westland, Rik et al. Kidney Int. 2015[13] | - | Y | TGA, VSD, COA, TOF, BAV, PA | Pierpont, Mary Ella et al. Circulation. 2018[16] CHDGKB database[17] | Y | Epilepsy, ID, Autism Spectrum Disorder | Borlot F et.al. JAMA Neurol. 2017[20] SFARI database[21] |
| 16p13 | del/dup | known CNV | KA (URA, MCDK) OU DCS | Sanna-Cherchi, Simone et al. Am J Hum Genet. 2012[3] Westland, Rik et al. Kidney Int. 2015[13] Verbitsky, Miguel et al. Nat Genet. 2019[4] | AXIN1, IFT140, TSC2, PKD1, PDPK1, MYH11 | N | - | - | Y | Autism Spectrum Disorder, Epilepsy, ID | SFARI database[21] Borlot F et.al. JAMA Neurol. 2017[20] |
| 17p12 | del/dup | known CNV | KA | Sanna-Cherchi, Simone et al. Am J Hum Genet. 2012[3] Verbitsky, Miguel et al. J Clin Invest. 2015[12] | - | N | - | - | N | - | - |
| 17p13 | del/dup | Likely pathogenic CNV known CNV | KA | Sanna-Cherchi, Simone et al. Am J Hum Genet. 2012[3] Verbitsky, Miguel et al. J Clin Invest. 2015[12] | - | N | - | - | N | - | - |
| 17q12 | del/dup | known CNV | KA PUV OU DCS Glomerular other | Sanna-Cherchi, Simone et al. Am J Hum Genet. 2012[3] Verbitsky, Miguel et al. J Clin Invest. 2015[12] | HNF1B | N | - | - | Y | Autism Spectrum Disorder | SFARI database[21] |
| 1q21 | del/dup | known CNV | KA VUR (Reflux nephropathy) PUV OU | Sanna-Cherchi, Simone et al. Am J Hum Genet. 2012[3] Verbitsky, Miguel et al. J Clin Invest. 2015[12] Verbitsky, Miguel et al. Nat Genet. 2019[4] | - | Y | PDA, VSD, ASD, TrA, TOF | Pierpont, Mary Ella et al. Circulation. 2018[16] CHDGKB database[17] | Y | Autism Spectrum Disorder, SCZ | SFARI database[21] Marshall CR et al. Nat Genet. 2017[19] |
| 1q44 | del | known CNV Likely pathogenic CNV | KA (URA) | Sanna-Cherchi, Simone et al. Am J Hum Genet. 2012[3] Westland, Rik et al. Kidney Int. 2015[13] | SDCCAG8, KIF26B, NLRP3 | Y | VSD, COA, HLHS | Pierpont, Mary Ella et al. Circulation. 2018[16] | N | - | - |
| 22q11 | del/dup | known CNV | KA PUV OU VUR DCS | Sanna-Cherchi, Simone et al. Am J Hum Genet. 2012[3] Verbitsky, Miguel et al. Nat Genet. 2019[4] | CRKL | Y | IAA-B, isolated aortic arch anomalies, TrA, TOF, HLHS, VSD, PVS | Pierpont, Mary Ella et al. Circulation. 2018[16] | Y | Autism Spectrum Disorder, SCZ, Epilepsy, ID | SFARI database[21] Marshall CR et al. Nat Genet. 2017[19] Borlot F et.al. JAMA Neurol. 2017[20] |
| 2p11 | del/dup | Likely pathogenic CNV | KA VUR (Reflux nephropathy) | Sanna-Cherchi, Simone et al. Am J Hum Genet. 2012[3] Verbitsky, Miguel et al. J Clin Invest. 2015[12] | - | N | - | - | N | - | - |
| 3p26 | dup | known CNV Likely pathogenic CNV | KA (MCDK) | Sanna-Cherchi, Simone et al. Am J Hum Genet. 2012[3] Westland, Rik et al. Kidney Int. 2015[13] | - | N | - | - | N | - | - |

**Table 1 (continued)**

| | CNV landscapes across CAKUT cases | | | | Extrarenal malformations (Cardiovascular) | | | | Extrarenal malformations (Neurocognitive) | | |
|---|---|---|---|---|---|---|---|---|---|---|---|
| CNV location | CNV type | Classification of CNV | Renal/urinary system phenotype CAKUT | Reference | Risk genes | Cardiovascular defect | Cardiovascular phenotype | Reference | Neurocognitive defect | Neurocognitive phenotype | Reference |
| 3q29 | del/dup | Likely pathogenic CNV / known CNV | KA (MCDK) | Sanna-Cherchi, Simone et al. Am J Hum Genet. 2012[3] Westland, Rik et al. Kidney Int. 2015[13] | DLG1 | N | - | - | Y | Autism Spectrum Disorder, SCZ | SFARI database[21] Marshall CR et al. Nat Genet. 2017[19] |
| 4p16 | del/dup | known CNV | KA | Sanna-Cherchi, Simone et al. Am J Hum Genet. 2012[3] Verbitsky, Miguel et al. J Clin Invest. 2015[12] Verbitsky, Miguel et al. Nat Genet. 2019[4] | FGFRL1, FGFR3, SLC2A9 | N | - | - | N | - | - |
| 7p21 | del/dup | known CNV / Intergenic CNV | KA (MCDK) | Sanna-Cherchi, Simone et al. Am J Hum Genet. 2012[3] Westland, Rik et al. Kidney Int. 2015[13] | - | N | - | - | N | - | - |
| 9q34 | dup | Intergenic CNV / Likely pathogenic CNV | KA | Sanna-Cherchi, Simone et al. Am J Hum Genet. 2012[3] Verbitsky, Miguel et al. J Clin Invest. 2015[12] | - | Y | ASD, VSD, TOF, pulmonary arterial stenosis | Pierpont, Mary Ella et al. Circulation. 2018[16] | N | - | - |

N No, Y Yes, del deletion, dup duplication, KA Kidney anomaly, PUV posterior urethral valve, OU obstructive uropathy, VUR vesicoureteral reflux, UPJO ureteropelvic junction obstruction, URA unilateral renal agenesis, MCDK multicystic dysplastic kidney, DCS duplication of the collecting system and/or ureter, COA coactation of aorta, TOF tetralogy of Fallot, TGA transposition of the great arteries, PDA patent ductus arteriosus, VSD ventricular septal defect, IAA-B interrupted aortic arch type B, ASD atrial septal defect, BAV bicuspid aortic valve, PA pulmonary atresia, HLHS hypoplastic left heart syndrome, TrA truncus arteriosus, TAPVR total anomalous pulmonary venous return, PVS pulmonic valve stenosis, SCZ Schizophrenia, ID Intellectual disability.

developing kidney (Kidney vs Heart: $P = 6.764 \times 10^{-16}$; Kidney vs Brain: $P = 7.254 \times 10^{-21}$; Kidney vs Cerebellum: $P = 3.125 \times 10^{-41}$; Fig. 6f and Supplementary Data 16).

## Discussion

As the leading cause of end-stage renal disease, CAKUT account for approximate 41.3% of children who receive renal replacement therapy[22]. Disturbances in the normal nephrogenesis can arise from exposure to environmental factors and defects in genes may lead to CAKUT. A better understanding of the genetic etiologies of CAKUT is essential for precision diagnosis, prevention, and treatment on CAKUT and its associated clinical outcomes. CNV has been recognized as a common cause of CAKUT. Sanna-Cherchi et al. found that the CNV distribution was significantly skewed toward larger gene-disrupting events in cases compared to ethnicity-matched controls ($P = 4.8 \times 10^{-11}$) through examining CNVs in 522 patients with renal hypodysplasia. 10.5% (55/522) of the cases harbored 34 CNVs associated with known genomic disorders, which was detected in only 0.2% of 13,839 controls ($P = 1.2 \times 10^{-58}$). Additionally, another 32 cases (6.1%) were found carrying large gene-disrupting CNVs that were absent from controls[3]; Verbitsky performed a genome-wide analysis of CNVs in 2824 CAKUT cases and 21498 controls, suggesting that CAKUT cases carried a significant burden of CNVs affecting coding genes and were enriched for known genomic disorders. Six loci including 1q21, 4p16.1-p16.3, 16p11.2, 16p13.11, 17q12, and 22q11.2 accounted for 65% of cases with CNVs associated with known genomic disorders[4]. In the present study, we summarized the recurrent CAKUT-associated CNVs and found that most of these CNVs (63.16%, 12/19) remained undetermined for the pathogenic genes (Table 1).

Like other congenital anomalies associated CNV studies, the etiologies of CAKUT-associated CNVs were currently interpreted based on their interference on the protein coding gene dosage. Despite the success of such strategy in several recurrent microdeletion syndromes such as the 17q21.31 deletion syndrome caused by haploinsufficiency of *KANSL1*[23], most CNVs also affect non-coding genomic regions that might have regulatory effect on the pathogenic mechanisms of diseases[5]. Recently, multiple lines of evidence indicated that CNVs might cause birth defects through affecting their encompassed lncRNAs[9], which have been identified as key regulators in development and disease[6,8,10]. In the present study, we found that CAKUT genes enriched modules (CAKUT_sig1 and CAKUT_sig2) were both negatively correlated with developmental stage (Fig. 2a–c and Supplementary Data 2). In previous studies over cross-species evolution of lncRNAs, Cardoso-Moreira and Sarropoulos et al. found that in contrast to most protein-coding genes (73–90% depending on the species), only a fraction of lncRNAs (16–38%) show developmentally dynamic expression and enrichment for functional loci[24,25]. In addition, across all organs, genes used early in development are less tolerant to loss-of-function mutations[24,25]. Taken together, these results support the relationship between the CAKUT_sig1 and CAKUT_sig2 modules and their enriched developmental functions (Fig. 2d and Supplementary Data 3), as well as potential contributions of lncRNAs within modules to the etiologies of CAKUT-associated CNVs. Noticeably, only a small fraction of protein coding genes in the CAKUT_sig1 (6.31%, 355/5630) and CAKUT_sig2 (9.78%, 22/225) modules, as well as known CAKUT genes (12.21%, 21/172, Supplementary Table 1) were distributed in 19 CAKUT-associated CNVs (Fig. 3a, Table 1 and Supplementary Data 2). In comparison, the CAKUT_sig1 (49.76%, 206/414) and CAKUT_sig2 (34.78%, 8/23) modules contained a considerable number of CNV-lncRNAs that highly correlated with CAKUT genes ($|r| \geq 0.7$, Fig. 3b and Supplementary Data 4, 5). In summary,

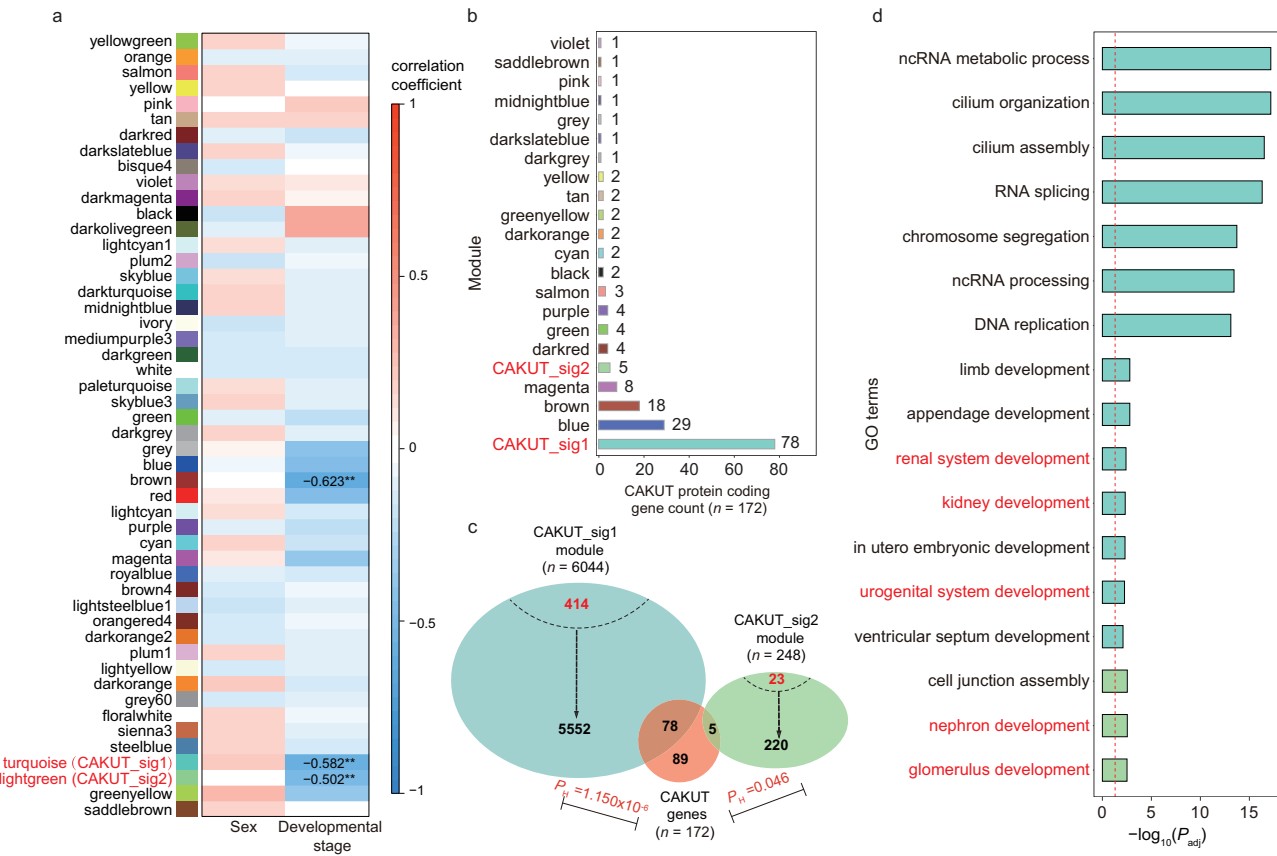

**Fig. 2 CNV-lncRNA developmental coexpression modules related to CAKUT and kidney development. a** Heatmap shows Pearson correlation coefficient (r) between 49 developmental coexpression modules and sample traits (Sex and developmental stage). Only significant correlation ($|r| \geq 0.5$, $P_{adj} < 0.05$) are labeled. The colors represent correlation coefficient value and direction (Supplementary Data 2). **b** Distribution of CAKUT genes ($n = 172$) in the developmental coexpression modules are shown in the bar plot. The y axis represents different CNV-lncRNA coexpression modules, and the x axis shows CAKUT gene counts in each module (Supplementary Data 2). **c** Two developmental stage-correlated modules that significantly enriched CAKUT gene sets ($P_H < 0.05$, $P_H$ represents hypergeometric P value) are shown (CAKUT_sig1 and CAKUT_sig2). Protein coding gene (including CAKUT gene) counts and CNV-lncRNA counts in each module are shown in black and red font, respectively. Significant enrichment is indicated with the $P_H$ value in red font. Protein coding genes that went for WGCNA ($n = 19957$) were used as the background gene list (Supplementary Data 2). **d** Functional annotations of two modules that enriched genes are shown. For each module, the top seven (ranked by $P_{adj}$) and organ development-associated functional annotations are shown (CAKUT_sig2 module only significantly enriched in three GO terms). GO terms that related to kidney development are in red font. The y axis represents GO terms, and the colors of the bars represent different CNV-lncRNA coexpression modules (Supplementary Data 3). The x axis displays the values of $-\log_{10}(P_{adj})$. The red dashed line indicates $P_{adj}$ of 0.05, ** denotes $P_{adj} < 0.01$ (Supplementary Data 1).

these results suggest that CAKUT-associated CNVs were more likely to affect the expression of CAKUT genes outside the CNV regions remotely by disturbing the lncRNAs within CNVs, thus contributing to CAKUT. We further validated this hypothesis through knockdown experiments of 2 hub CNV-lncRNAs *HSALNG0134318* and *HSALNG0115943* in HEK293 cell lines (Fig. 4a–c and Supplementary Data 7, 8).

Most of the CNV-lncRNAs are not evolutionarily conserved[25]. Therefore, compared to traditional animal models, the kidney organoids have become an ideal in vitro model to study the molecular mechanism of human kidney development[26,27]. Single cell RNA-seq analyses have supported the fidelity of kidney organoids as models of the developing kidney and affirmed their potential in disease modeling[28]. Based on the RNA-seq data of in vitro kidney organoids differentiation, we verified the coexpression between CAKUT genes and 2 hub CNV-lncRNAs *HSALNG0134318* and *HSALNG0115943* (Fig. 5b, c and Supplementary Data 9). Surprisingly, *HSALNG0134318* and *HSALNG0115943* were hub CNV-lncRNAs (MM ≥ 0.8) in both kidney developmental WGCNA and kidney organoids differentiation WGCNA (Figs. 3a, 5a and Supplementary Data 2, 9). In

addition, kidney organoids differentiation modules of *HSALNG0134318* and *HSALNG0115943* were also related to kidney and embryonic development (Fig. 5a, d and Supplementary Data 9). Furthermore, kidney organoids have overcome the limitations of two-dimensional cell cultures, whereby branching of the ureteric bud and formation of cellularly complex three-dimensional structures were unattainable[27]. To specifically study the expression patterns of *HSALNG0134318* and *HSALNG0115943* in different kidney tissues, we characterized their differential expression patterns between different kidney organoid-derived tissues (glomeruli and proximal tubules), which provided initial clues for further studies on their molecular mechanisms in CAKUT (Fig. 5e and Supplementary Data 10).

It's worth pointing out that *CRKL* has been identified as the main genetic driver of kidney defects in 22q11.2 CNV[29,30]. In fact, *CRKL* locates at the central part of 22q11.2 (22q11.21), and *HSALNG0134318* locates at the distal part of 22q11.2 (22q11.22, chr22:22,298,141-22,307,554, hg38). Since the occurrence of kidney abnormalities in 22q11.2 CNV that did not involve *CRKL* was still unexplained, the genetic effect of other candidates in 22q11.2 region should not be ignored. It's worth noting that

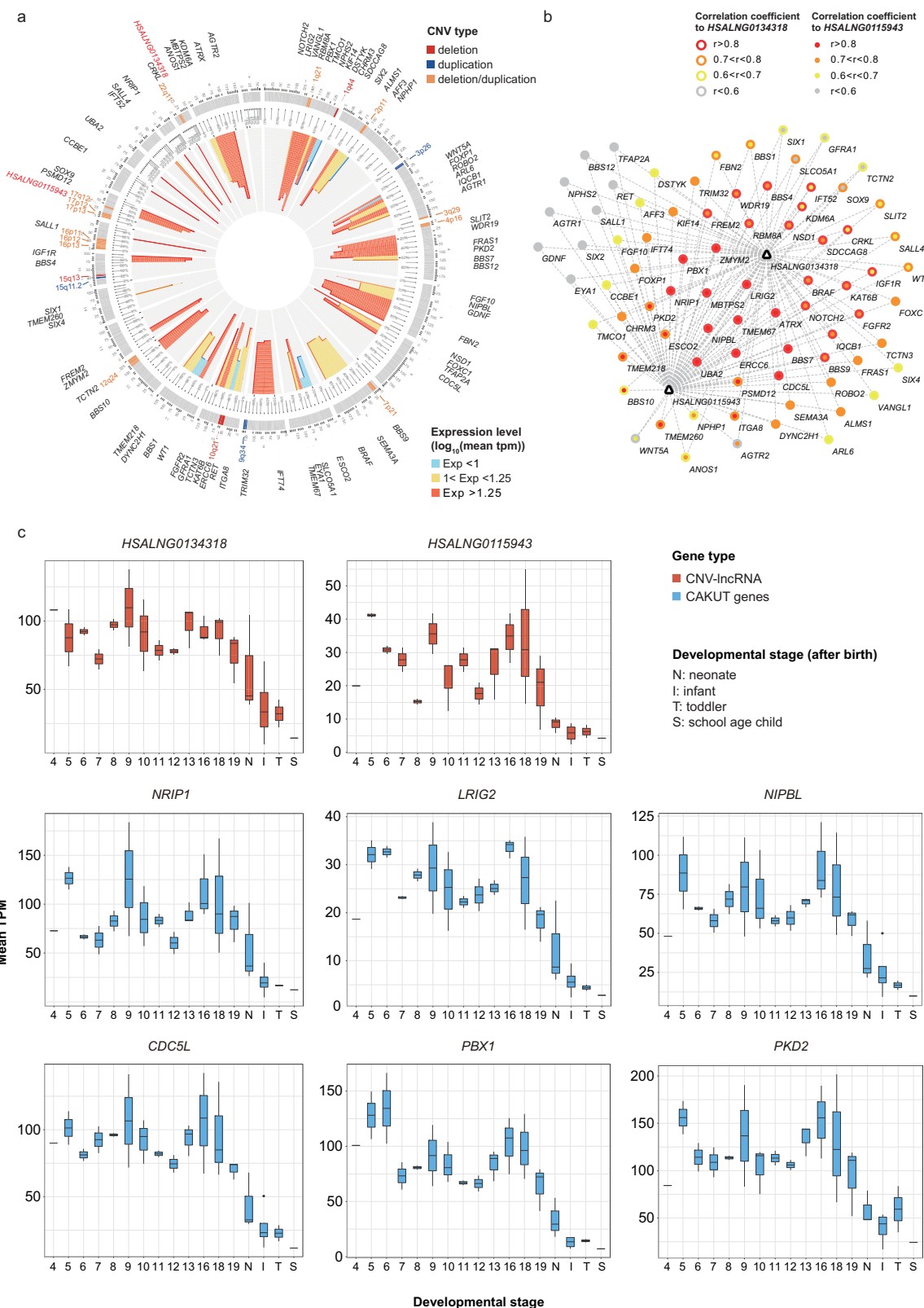

expression of *HSALNG0134318* and *CRKL* showed highly correlated expression in developmental kidneys (Fig. 3b), and knockdown of *HSALNG0134318* would significantly downregulate the expression of *CRKL* ($P = 3.778 \times 10^{-11}$, Fig. 4a). Therefore, expression disturbance of *CRKL* could be the downstream effect of *HSALNG0134318* deletion other than the dosage effect of *CRKL* itself. We identify 4 driver lncRNAs regulating CAKUT

genes through ceRNA mechanism in the CAKUT_sig1 and CAKUT_sig2 modules using the LncmiRSRN[31] (Supplementary Fig. 5). *HSALNG0134318* was not identified as a driver lncRNA since only top 20% of tested lncRNAs with enriched targeted miRNAs were selected in this algorithm. Previously, miRNA miR-185 located in 22q11.2 was shown to be associated with CAKUT[32]. Therefore, we further investigated the potential

**Fig. 3 Hub CNV-lncRNAs *HSALNG0134318* and *HSALNG0115943* with multiple coexpressed CAKUT genes during kidney development. a** The circos plot shows CAKUT-associated CNVs ($n = 19$), two hub CNV-lncRNAs (*HSALNG0134318*: MM = 0.87; *HSALNG0115943*: MM = 0.85) and coexpressed CAKUT genes ($n = 78$) in the CAKUT_sig1 module. The colors of CNVs represent the types of CNV. Two hub CNV-lncRNAs are highlighted in red font. The height of bars in the inner circos represent the mean expression level (TPM) of corresponding genes during kidney development (Supplementary Data 1, 4). **b** Pearson correlation coefficient between two hub CNV-lncRNAs (*HSALNG0134318* and *HSALNG0115943*) and coexpressed CAKUT genes ($n = 78$) in the CAKUT_sig1 module during kidney development are shown. The colors of inner and outer circles represent Pearson correlation coefficient of CAKUT genes to *HSALNG0115943* and *HSALNG0134318*, respectively (Supplementary Data 4, 5). **c** Expression patterns of two hub CNV-lncRNAs (*HSALNG0134318* and *HSALNG0115943*, in red boxes) and predicted regulated CAKUT genes (in blue boxes) are shown (Supplementary Data 6). The numbers on x axis represents embryonic developmental stage (week). The y axis represents mean expression value of each stage. The center line represents a median value. The box limits represent upper and lower quartiles. The whiskers represent 1.5x interquartile range. The points represent outliers. Data in Fig. 4 were calculated using the human kidney developmental dataset ($n = 40$). MM, module membership; r, Pearson correlation coefficient (Supplementary Data 1).

interaction of lncRNAs and miRNAs in 22q11.2. *HSALNG0134318* is also the target of two miRNAs (has-miR-3198 and has-miR-1306-5p), of which has-miR-3198 also target the CAKUT gene *DHODH* (Supplementary Fig. 6 and Supplementary Data 12). Whereas, knockdown of *HSALNG0134318* had no effect on the expression of *DHODH* (Supplementary Data 8). The miRNA miR-185 could targets 22q11.2 CNV-lncRNAs (*HSALNG0134144* and *HSALNG0134523*) and CAKUT genes (*ATN1*, *EMC10*, *GNB2*, *IGF1R*, *NSD1*, *SIX1* and *SIX5*) (Supplementary Fig. 6 and Supplementary Data 12). We also noticed that three CAKUT genes including *IFG1R* (log$_2$FoldChange = −0.564, $P_{adj} = 3.413 \times 10^{-5}$), *IFT52* (log$_2$FoldChange = 0.747, $P_{adj} = 2.282 \times 10^{-4}$), and *CRKL* (log$_2$FoldChange = −0.569, $P_{adj} = 1.364 \times 10^{-4}$), which could be significantly regulated through knockdown of *HSALNG0134318*, were targets of miRNAs located at 22q11.2 (Supplementary Fig. 6 and Supplementary Data 8). Therefore, 22q11.2 ncRNAs and mRNAs could have connections during kidney development. Deletion of such region should have synergy effect on clinical phenotypes. Since knockdown of *HSALNG0134318* affected the expression of *CRKL*, *HSALNG0134318* could be upstream of the kidney developmental pathways involving *CRKL*. Through transcription factor prediction, we identified the CAKUT gene *TFAP2A*, encoding a transcription factor, could govern the expression of *HSALNG0134318* in kidney development (Supplementary Table 2 and Supplementary Data 12).

The complexity of birth defects in patients with syndrome increases the difficulty of clinical diagnosis, treatment, and management. CAKUT, CHD, and NDDs have been found to share pathogenetic CNVs[33]. The current study also identified overlapping recurrent CNVs and known genes associated with CAKUT, CHD and NDDs (Table 1, Supplementary Fig. 1, Fig. 6d). The CAKUT_sig1 module contained the largest amounts of extrarenal malformation-causing CAKUT genes (43.8%, 14/32, Fig. 6e; Supplementary Table 3). Since the CAKUT_sig1 module was also involved in multiple phylogenetic functions (Fig. 2d and Supplementary Data 3), we further compared the expression patterns of its CNV-lncRNAs and protein coding genes in the kidney, heart, brain, and cerebellum developmental samples. We found that the kidney-biased genes were associated with development of multiple organs, including kidney, neural, cardiac and other organ development (Fig. 6c, f and Supplementary Data 13–16). Regarding the potential mechanisms underlying the involvement of the CAKUT_sig1 module in multiple phylogenetic disorders, functional enrichment analyses indicated that the most significant (ranked by $P_{adj}$) functions of the CAKUT_sig1 module involved ncRNA metabolic process ($P_{adj} = 6.506 \times 10^{-18}$) and ncRNA processing ($P_{adj} = 3.666 \times 10^{-14}$) (Fig. 2d and Supplementary Data 3). This highlights the potential roles of CNV-lncRNAs in this module. On the other hand, the most significant functions also involved cilium organization ($P_{adj} = 6.506 \times 10^{-18}$) and cilium assembly ($P_{adj} = 3.022 \times 10^{-17}$) (Fig. 2d and Supplementary Data 3). The cilia determine left-right axis patterning

during embryogenesis and have been reported to play an important role in cardiac, renal, limb, liver and spleen development[34,35]. In view of the developmental pathways involved in the CAKUT_sig1 module were mostly affected by left-right axis development that determined in early embryonic stage (Fig. 2d and Supplementary Data 3)[34], we speculate that the cilia assembly process regulated by CNV-lncRNAs could potentially be related to multiple phylogenetic disorders, which is worthy of further study.

There are several limitations in this study that could be addressed in future research. First, based on human kidney developmental and kidney organoids differentiation transcriptome datasets, we have proven stage-specific expression and strong driving roles of CNV-lncRNAs during nephrogenesis at both organ and cellular levels. However, the physiological and molecular mechanisms in which CNV-lncRNAs are involved have not been elucidated. Similarly, knockdown experiments of 2 hub CNV-lncRNAs *HSALNG0134318* and *HSALNG0115943* in HEK293 cell lines provided evidences that they could regulate several known CAKUT genes including *CRKL* (Fig. 4a, b and Supplementary Data 7), as well as affect pathways involved in kidney and multiple systems development (Fig. 4c, Supplementary Fig. 2d and Supplementary Data 8). Nevertheless, a lack of deep insight over these mechanisms is a limitation of the study. Since lncRNAs can either repress or activate gene expressions in *cis*, in *trans* or even through epigenetic modification[6,36], iPSC-induced kidney organoids, cardiomyocytes and neural progenitor cells in vitro models combined with high-throughput sequencing technologies such as ChIP-seq, scRNA-seq, and proteomics could be further used to study the molecular mechanisms by which *HSALNG0134318* and *HSALNG0115943* cause CAKUT and concomitant extrarenal malformation. Secondly, the significance of enrichment in known CAKUT gene sets was related to gene counts in each module and the size of known CAKUT gene sets. Since CAKUT genes that have been identified so far is limited (totaled to 172 CAKUT genes in current study, Supplementary Table 1), modules other than the CAKUT_sig1 and CAKUT_sig2 modules (e.g., the kidney developmental brown module that contained 18 CAKUT genes, $P = 0.573$; the kidney developmental magenta module that contained 8 CAKUT genes, $P = 0.076$) could not be ignored in the pathogenesis of CAKUT (Fig. 2b, c and Supplementary Data 2). Lastly, the accumulation of more clinical genetic data, especially non-coding genomic variants data, and comprehensive phenotypic information of patients is needed to help us identify more CAKUT-associated CNV-lncRNAs. Altogether, we still have a long way to go to provide substantial insight into the regulatory roles of CNV-lncRNAs in CAKUT and extrarenal malformations.

In conclusion, we revealed potential regulatory relationships between CNV-lncRNAs and CAKUT genes in human kidney developmental and kidney organoids differentiation transcriptome datasets. For the CAKUT_sig1 module that most

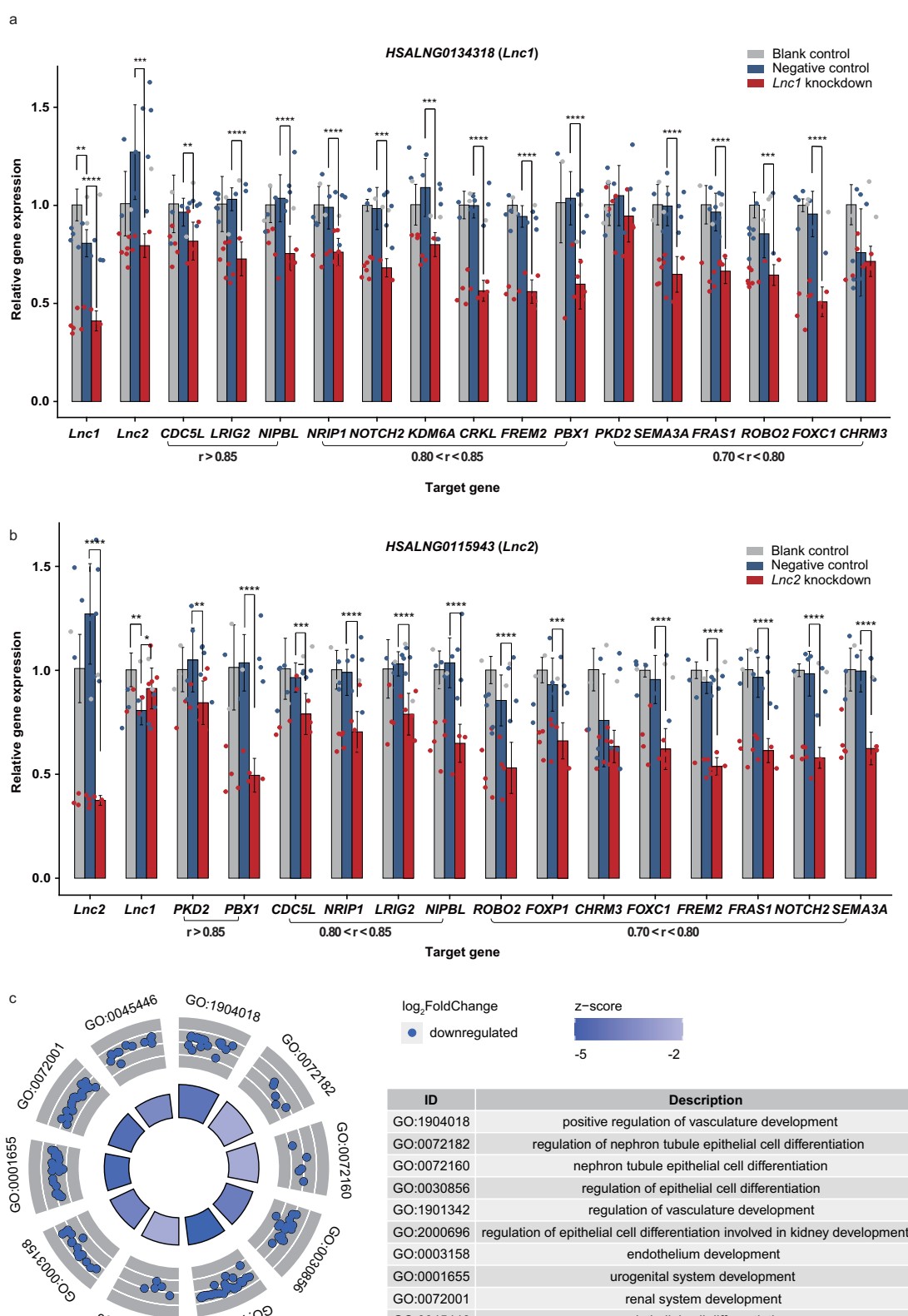

enriched CAKUT genes, we performed in vitro validation of two hub CNV-lncRNAs *HSALNG0134318* and *HSALNG0115943* located in well-known CAKUT associated CNVs 22q11.2 and 17q12 respectively. The results suggested that *HSALNG0134318* and *HSALNG0115943* play strong regulatory roles on the CAKUT genes and pathways involved in kidney and multi-organ development. Our results highlighted the potential contribution of

lncRNAs within CNVs to the pathogenetic process of CAKUT and concomitant extrarenal malformation.

## Methods

**Study design.** We aimed to investigate the potential contribution of lncRNAs to the pathogenicity of CAKUT associated CNVs. 19 recurrent CAKUT-associated CNVs were identified based on clinical researches of congenital anomalies of the

**Fig. 4 Reduction of hub CNV-lncRNAs may affect CAKUT gene expression and kidney development. a, b** qPCR analyses were used to detect expression changes for CAKUT genes that were coexpressed with two hub CNV-lncRNAs after knockdown of CNV-lncRNA *HSALNG0134318* (*Lnc1*, **a**) and *HSALNG0115943* (*Lnc2*, **b**) in human embryonic kidney cell lines (HEK293). The colors of bars and dots represent different treatment groups. Pearson correlation coefficients of CAKUT genes to CNV-lncRNA in the human kidney developmental dataset ($n = 40$) are labeled below. The knockdown experiments were conducted in at least three biological replicates. The error bars are shown as means ± SD. Two-tailed $t$ test was used for comparison between two groups (Supplementary Data 7). **c** The GOcircle plot shows enriched kidney development-associated pathways of down-regulated genes ($\log_2$FoldChange < 0 and $P_{adj} < 0.05$, $n = 839$) after *HSALNG0134318* knockdown in the HEK293 cell line are shown in GOcircle plot. The colors of bars in the inner circle represent z-score of each GO term. Distributions and colors of dots in the outer circle represent $\log_2$FoldChange of genes enriched in each GO term. Descriptions of enriched GO terms are shown in the right panel (Supplementary Data 8). Significant levels are indicated as follows: *$P < 0.05$, **$P < 0.01$, ***$P < 0.001$, ****$P < 0.0001$ (Supplementary Data 1).

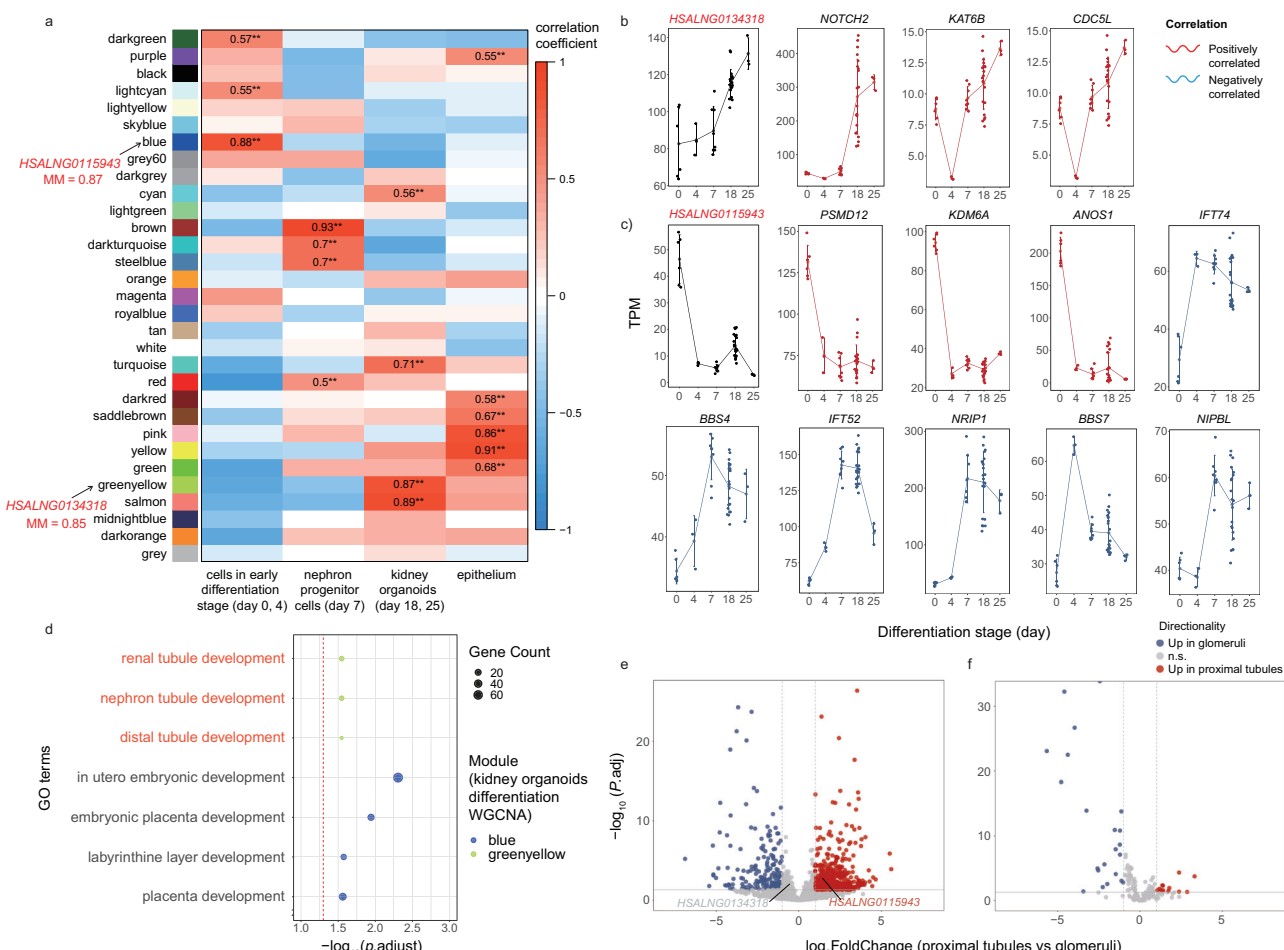

**Fig. 5 Driving roles and characteristics of hub CNV-lncRNAs *HSALNG0134318* and *HSALNG0115943* during kidney organoid differentiation. a** The heatmap shows Pearson correlation coefficient (r) between 30 kidney organoids coexpression modules and sample traits (differentiation stage). Modules and corresponding module membership (MM) values of two hub CNV-lncRNAs (*HSALNG0134318* and *HSALNG0115943*) are marked on the left. Only significant correlation ($|r| \geq 0.5$, $P_{adj} < 0.05$) are labeled. The colors represent correlation coefficient value and direction (Supplementary Data 9). **b, c** Expression patterns of two hub CNV-lncRNAs (*HSALNG0134318*, **b** and *HSALNG0115943*, **c**) and predicted regulated CAKUT genes, which were coexpressed both in developmental WGCNA and kidney organoids WGCNA, are shown in line plots (Supplementary Data 9). Positively and negatively correlated CAKUT genes are represented by red and blue lines, respectively. The $x$ axis represents kidney organoids differentiation stages from iPSCs to mature kidney organoids. The $y$ axis represents mean expression value (TPM) of each stage. The error bars show mean ± SD of each stage. **d** Functional annotations of kidney organoids coexpression modules that contain two hub CNV-lncRNAs (*HSALNG0134318*: greenyellow, *HSALNG0115943*: blue) are shown. For each module, organ development-associated functional annotations are displayed. The $x$ and $y$ axes represent values of $-\log_{10}(P_{adj})$ and GO terms, respectively. The red dashed line indicates $P_{adj}$ of 0.05. The colors of the dots represent different modules. The sizes of the dots represent enriched gene counts in each GO terms. GO terms that directly related to kidney development are in red font (Supplementary Data 9). **e, f** Differential expression patterns of CNV-lncRNAs (**e**, $n = 8997$) and CAKUT genes (**f**, $n = 172$) between mature kidney organoid derived proximal tubules ($n = 3$) and glomeruli ($n = 6$) are shown in volcano plots (Supplementary Data 9). Two hub CNV-lncRNAs (*HSALNG0134318* and *HSALNG0115943*) are labeled in red font. The $x$ and $y$ axes represent $\log_2$FoldChange (proximal tubules vs glomeruli) and $-\log_{10}(P_{adj})$, respectively. Red dots represent significantly up-regulated genes in the proximal tubules ($\log_2$FoldChange $\geq 1$, $P_{adj} < 0.05$). Blue dots represent significantly up-regulated genes in the glomeruli ($\log_2$FoldChange $\leq -1$, $P_{adj} < 0.05$). Gray dots represent genes that do not show differential expression. The horizontal and vertical red dashed line indicate $P_{adj} = 0.05$ and $|\log_2$FoldChange$| = 1$, respectively, ** denotes $P_{adj} < 0.01$ (Supplementary Data 1).

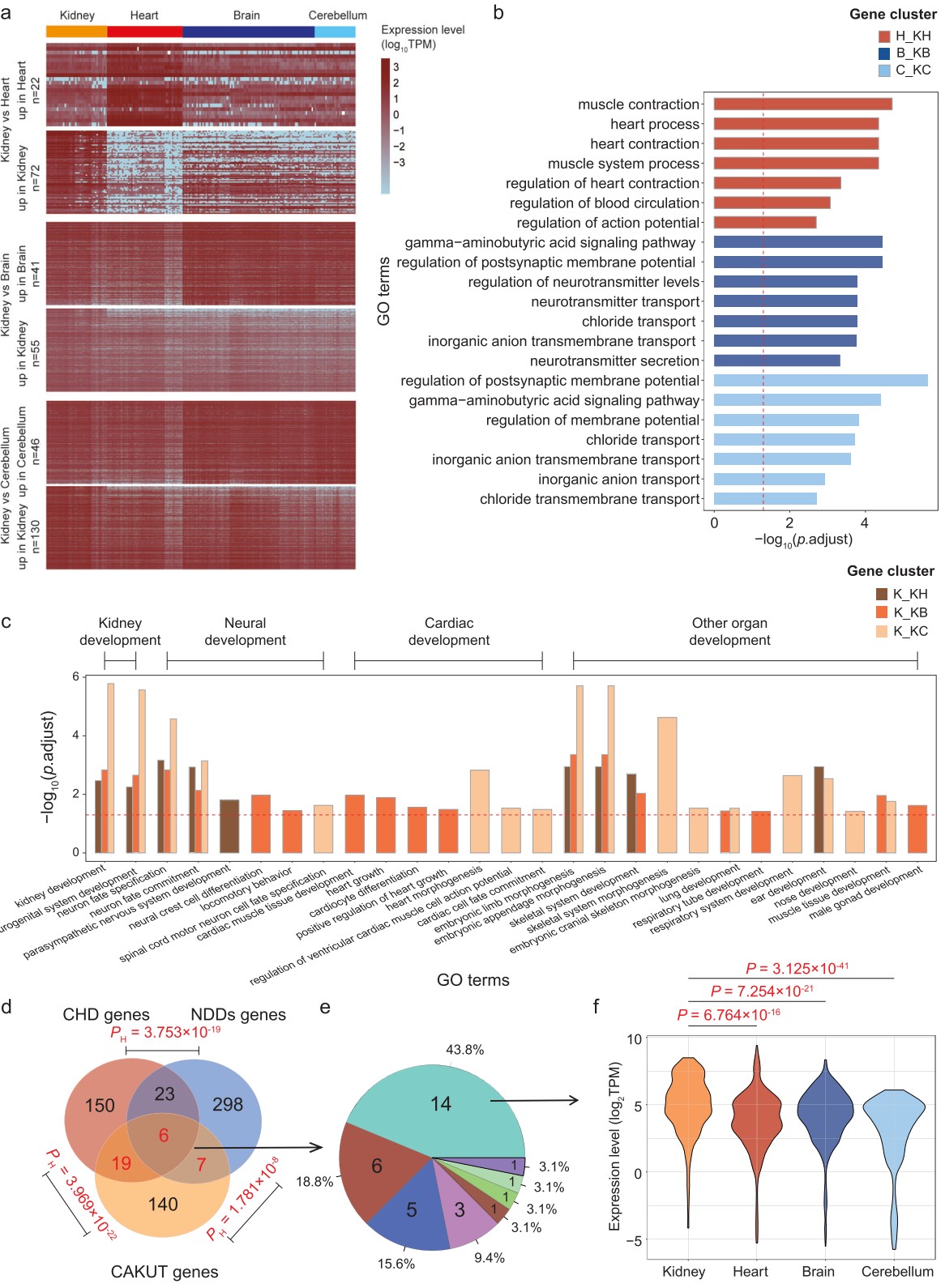

kidney and urinary tracts (CAKUT) cases[3,4,12,13] (Table 1). We retrieved lncRNAs located within these genomic regions as candidate CNV-lncRNAs (Supplementary Data 2). To establish the kidney developmental coexpression networks for the CNV-lncRNAs, we performed weighted gene coexpression network analysis (WGCNA) over candidate CNV-lncRNAs and protein coding genes using the human kidney developmental RNA-seq data (*n* = 40) from LncExpDB[11] (Supplementary Data 2). Downstream analyses, including pathways analyses and enrichment analyses for known CAKUT genes, were used to

identify coexpression modules associated with kidney development and CAKUT. We mainly focused on the CAKUT genes enriched CAKUT_sig1 module, which showed significant correlation (|r| ≥ 0.5, $P_{adj}$ < 0.05) with developmental stage and multi-organ development including kidney and urinary tract ($P_{adj}$ < 0.05). To explore the molecular basis of underlying extrarenal malformation caused by CAKUT associated CNVs (Table 1), we performed multiple organs differential expression analyses over CNV-lncRNAs and protein coding genes in the CAKUT_sig1 module.

**Fig. 6 Differentially expressed and multifunctional genes in the CAKUT_sig1 module. a** For the CAKUT_sig1 module, the heatmap shows the expression landscape of significantly differentially expressed CNV-lncRNAs and protein coding genes ($|\log_2\text{FoldChange}| \geq 4$, $P_{adj} < 0.05$) in the developmental kidney, heart, brain and cerebellum (Kidney samples: $n = 40$, heart samples: $n = 50$, brain samples: $n = 87$, cerebellum samples: $n = 27$). The counts (including CNV-lncRNAs and protein coding genes) of differentially expressed genes in each cluster are shown on the left. The tissue types of corresponding samples are labeled above the heatmap. The colors represent gene expression value ($\log_{10}$TPM) in different samples (Supplementary Data 13–15). **b** Functional annotations of heart, brain and cerebellum differentially expressed gene clusters compared to kidney samples are shown (Supplementary Data 13–15). For each gene cluster, the top seven biological process GO terms (ranked by $P_{adj}$) are shown. The x and y axes represent values of $-\log_{10}(P_{adj})$ and GO terms, respectively. The red dashed line indicates $P_{adj}$ of 0.05. The colors of the bars represent different gene clusters. **c** Functional annotations of kidney differentially expressed gene clusters compared to heart, brain and cerebellum samples are shown (Supplementary Data 13–15). For each gene cluster, organ development-associated functional annotations are shown. The associated organs of corresponding GO terms are labeled above the bar plot. The x and y axes represent GO terms and values of $-\log_{10}(P_{adj})$, respectively. The red dashed line indicates $P_{adj}$ of 0.05. The colors of the bars represent different gene clusters. **d** Gene counts and intersection of congenital heart disease (CHD), neurodevelopmental disorders (NDDs) and CAKUT gene sets are shown in Venn diagrams (Supplementary Data 16). Intersection counts are shown in red font. **e** Distribution of known CAKUT genes associated with two or more phylogenetic disorders (intersection in **d**, $n = 32$) in developmental WGCNA modules (Supplementary Table 3; Supplementary Data 16). **f** Expression patterns of CAKUT genes associated with extrarenal development defects in the CAKUT_sig1 module ($n = 14$) in different tissues are shown in the violin plots. The x and y axes represent tissue types and gene expression levels ($\log_2$TPM), respectively (Supplementary Data 1, 16). CHD congenital heart disease, NDD neurodevelopmental disorders, CAKUT congenital anomalies of the kidney and urinary tracts.

---

Two hub CNV-lncRNAs in the CAKUT_sig1 module, *HSALNG0134318* within 22q11.2 and *HSALNG0115943* within 17q12, showed strong coexpression and correlation with multiple key CAKUT genes. Using the in vitro kidney organoids differentiation RNA-seq data ($n = 53$), we validated the key driving roles of *HSALNG0134318* and *HSALNG0115943* during the formation of kidney structure. To characterize the expression patterns of *HSALNG0134318* and *HSALNG0115943* in the specific kidney tissues, differential expression analyses were conducted between the kidney organoids derived proximal tubules and glomeruli.

To further confirm the predicted regulatory effects of *HSALNG0134318* and *HSALNG0115943* on CAKUT genes expression and kidney development, in vitro knockdown experiments along with qPCR analyses and RNA-seq analyses were performed in the human embryonic (HEK293) kidney cell lines.

**Human kidney developmental data**. The gene expression data from LncExpDB[11] were generated through comprehensive annotation for lncRNAs of the original dataset of human kidney developmental samples ($n = 40$) collected from ArrayExpress (E-MTAB-6814) (Supplementary Data 2). The tissue sampling of embryonic samples ($n = 30$) started at 4 weeks post-conception and continued weekly until 20-week post-conception except for 14-, 15-, and 17-week post-conception. Postnatal tissues ($n = 10$) were sampled from neonates, infants (6 ~ 9 months), toddlers (2 ~ 4 years), and school age children (7 ~ 8 years)[25].

**In vitro kidney organoids differentiation RNA-seq data**. To validate the roles of CNV-lncRNAs in the formation of kidney structure, we collected a series of RNA-seq datasets that covered several differentiation stages (days 0, 4, 7, 18, 25, and 28) from human induced pluripotent stem cells (iPSCs) to kidney organoids. Candidate samples were differentiated from human iPSC line in the same laboratory[37,38], following the same growth protocol and sequencing platform (GPL18573, Illumina NextSeq 500). RNAs were extracted using QIAGEN RNeasy micro kit. RNA libraries were prepared for sequencing using standard Illumina protocols. Only samples of wild genotype were included in our study to avoid biases caused by gene expression interference. The number of biological replicates of each differentiation stage ranges from 3 to 21, for a total of 41 single-end RNA-seq libraries and 12 paired-end RNA-seq libraries. For single-end RNA-seq samples ($n = 41$), the raw data were available from the Gene Expression Omnibus (GEO) under accession GSE88942, GSE89044, GSE99468, GSE99469, and GSE107305. For paired-end RNA-seq samples ($n = 12$), the raw data were available from the Gene Expression Omnibus (GEO) under accession GSE99583, GSE99582, and GSE111992. Detailed information and growth protocols of the kidney organoids were available in the GEO accessions.

**Human heart, brain, and cerebellum developmental data**. The annotated human heart ($n = 50$), brain ($n = 87$) and cerebellum ($n = 27$) developmental data from LncExpDB[11] originated from ArrayExpress (E-MTAB-6814). The tissue sampling started at 4 weeks post-conception and continued weekly until 20-week post-conception except for 14-, 15-, and 17-week post-conception. Postnatal tissues were sampled from neonates, infants (6 ~ 9 months), toddlers (2 ~ 4 years), school age (7 ~ 9 years), teenagers (13 ~ 19 years), and adults (~65 years)[25].

**Known CAKUT, CHD, and NDDs gene sets**. Known CAKUT genes were summarized from the Online Mendelian Inheritance in Man database (https://www.omim.org)[39], as well as from reviews on CAKUT[1,33,40,41] (Supplementary Table 1). The known congenital heart disease (CHD) gene set was generated in our previous study[10]. For known neurodevelopmental disorders (NDDs), genes were summarized from studies including autism spectrum disorder[21] (ASD, $n = 213$),

schizophrenia[42,43] (SCZ, $n = 76$), intellectual disability[44] (ID, $n = 45$) and epileptic encephalopathies[45,46] (EE, $n = 46$). After removing duplicates, a total of 172 CAKUT, 198 CHD and 334 NDDs genes were included in multiple organ development analyses.

**Recurrent CAKUT associated CNVs and CNV-lncRNAs retrieval**. Nineteen recurrent CAKUT-associated CNVs (pathogenic or likely pathogenic CNVs identified in at least two CAKUT cohorts) were identified based on four independent clinical studies, totaling 2824 CAKUT cases[3,4,12,13] (Table 1). The annotated lncRNAs[11,15] that mapped to the genomic regions of 19 recurrent CAKUT associated CNVs were retrieved using BEDTools v2.29.2[47].

**WGCNA of human kidney developmental data**. Normalized gene expression data (TPM) involving candidate genes from the human kidney developmental RNA-seq data were used for coexpression analysis. The R package WGCNA v1.70[48] was used for coexpression network construction and module identification. Pearson correlation was used to calculate module-trait correlations. For module-trait correlations, adjusted $P$ value was caluculated with corPvalueStudent function in WGCNA R package. The power value was set at three for coexpression network construction. The parameters for the blockwiseModules function were as follows: maxBlockSize=5000, TOMType = "unsigned", minModuleSize = 30, reassignThreshold = 0, mergeCutHeight = 0.25, numericLabels = T and pamRespectsDendro = F. The module membership (MM) value was used to estimate the correlation between each gene/lncRNA and the module eigengene of a given module. The CNV-lncRNAs in each module with MM values ≥ 0.8 were defined as hub CNV-lncRNAs[8]. Protein coding genes in each module were used as the input for Gene ontology (GO) enrichment analysis and Kyoto Encyclopedia of Genes and Genomes (KEGG) pathway enrichment analyses with the R package clusterProfiler v3.10[49].

**Multiple organs differential expression analyses for the kidney developmental CAKUT_sig1 module**. To identify the molecular basis underlying extrarenal malformation caused by CAKUT-associated CNVs, particularly cardiovascular defects and neural developmental disorders (Table 1), we performed differential expression analyses over CNV-lncRNAs and protein coding genes in the CAKUT_sig1 module (significantly correlated to developmental stage and multiple organs development including kidney and urinary tract, Fig. 2). Human kidney, heart, brain and cerebellum developmental data were used for differential expression analyses (see 'Human kidney developmental data' and 'Human heart, brain and cerebellum developmental data' section). Totally three groups (kidney vs heart, kidney vs brain and kidney vs cerebellum) of differential expression analyses were conducted using DESeq2 v1.32[50], with six sets of differential expressed genes identified (Fig. 5). The cut-off values were set as the adjusted $P < 0.05$ and the $\log_2$fold change ≥4. Protein coding genes and CAKUT genes in each gene sets were used as the input for Gene ontology (GO) enrichment analysis and Kyoto Encyclopedia of Genes and Genomes (KEGG) pathway enrichment analyses with the R package clusterProfiler v3.10[49].

**Quantification of lncRNA and CAKUT genes expression levels in the kidney organoids**. The raw data (fastq files) of in vitro kidney organoids differentiation RNA-seq data ($n = 53$) were utilized to generate read counts for the set of lncRNAs and protein-coding genes. We mapped the reads from each library against the reference genome from LncBook[15], including well annotated lncRNAs ($n = 101293$) and protein-coding genes ($n = 19957$). The data were processed as follows: quality analyses and base quality filtering with FastQC v0.11.9 (https://www.bioinformatics.

babraham.ac.uk/projects/fastqc) and Trim Galore v0.6.6 (https://github.com/FelixKrueger/TrimGalore), rRNA removing with SortMeRNA v4.3.4[51], alignment with STAR v2.7.10[52], reads counting with featureCounts[53] implemented in the Subread package v2.0.3. We normalized read counts using TPM and then identified CNV-lncRNAs and CAKUT genes expression levels in each sample.

**WGCNA of in vitro kidney organoids differentiation RNA-seq data**. Genes that did not reach expression level of 10 TPM in any sample were filtered. With this screening criteria, 621 CNV-lncRNAs (6.90%, 621/8997) and 12087 protein-coding genes (60.57%, 12087/19957) were selected for WGCNA in the in vitro kidney organoids differentiation RNA-seq data. The normalized gene expression data (TPM) involving candidate genes for the in vitro kidney organoids differentiation RNA-seq data were used for coexpression analysis. The power value was set at 14 for coexpression network construction. Other parameters involved in kidney organoids WGCNA were the same as developmental WGCNA (see 'WGCNA of human kidney developmental data' section).

**Characterized of differentially expressed CNV-lncRNAs and protein-coding genes in the proximal tubules and glomeruli**. Using paired-end RNA-seq samples (day 25) (see 'In vitro kidney organoids differentiation RNA-seq data' section), we identified the CNV-lncRNAs ($n = 8997$) and protein-coding genes ($n = 19957$) that differentially expressed between mature kidney organoid derived proximal tubules ($n = 3$) and glomeruli ($n = 6$) using DESeq2 v1.32[50]. The cut-off values were set as the adjusted $P < 0.05$ and $\log_2$fold change $\geq 1$. To identify the functions of proximal tubules and glomeruli differentially expressed genes, protein coding genes in each differentially expressed gene sets were used as the input for Gene ontology (GO) enrichment analysis and Kyoto Encyclopedia of Genes and Genomes (KEGG) pathway enrichment analyses with the R package clusterProfiler v3.10[49].

**Pairwise gene expression correlations between two hub CNV-lncRNAs and CAKUT genes**. To identify reliable correlations between two hub CNV-lncRNAs (HSALNG0134318 and HSALNG0115943) and CAKUT genes ($n = 172$) during kidney development and kidney organoids differentiation, three datasets that cover different development and differentiation stages were used to calculate pairwise gene expression correlations: Human kidney developmental data ($n = 40$), single-end in vitro kidney organoids differentiation data ($n = 41$, day 0–25) and paired-end in vitro kidney organoids differentiation data ($n = 12$, day 25–28) (see 'Human kidney developmental data' and 'In vitro kidney organoids differentiation RNA-seq data' section). For each dataset, Pearson correlation was employed to calculate pairwise gene expression correlations. The $P$ values were adjusted for multiple testing using the Benjamini–Hochberg method. The cut-off of significant correlation was set as absolute value of correlation coefficient (r) $\geq 0.5$ and the adjusted $P < 0.05$.

**CNV-lncRNAs knockdown in human embryonic kidney (HEK293) cell lines**. Knockdown experiments of HSALNG0134318 and HSALNG0115943 were conducted through transient transfection with Smart Silencers (mixture of small interfering RNAs and antisense oligonucleotides targeting CNV-lncRNAs, synthesized by RiboBio, Guang Zhou, China). The Smart Silencer was transfected into HEK293 cell lines (RRID: CVCL_0045, Cyagen, Suzhou, China) with the Lipofectamine RNAiMAX (Invitrogen, Carlsbad, CA, USA). The silencer sequences were listed in Supplementary Table 4.

**Quantitative reverse transcription qPCR analyses**. A total of 1 μg cellular RNA was used as the template for cDNA preparation with the PrimeScript RT Reagent Kit (Takara, Dalian, China). Quantitative reverse transcription qPCR (RT-qPCR) was performed with the TB Green Premix Ex Taq II kit (Takara, Dalian, China) on the CFX 96 Real-Time PCR detection system (Bio-Rad Laboratories, Inc., Hercules, CA, United States). Relative gene expression levels were calculated based on the $2^{-\triangle\triangle Ct}$ method. At least three biologically independent experiments were conducted for each group. GAPDH was used as the internal reference gene. The RT-qPCR primer pairs were listed in Supplementary Table 5.

**RNA-seq analyses of HSALNG0134318 knockdown HEK293 cell lines**. For HSALNG0134318 knockdown HEK293 cells ($n = 3$) and control groups ($n = 3$), RNA was harvested using NovaSeq 6000 Reagent Kit. 2 ug of total RNA was used for the construction of sequencing libraries. RNA libraries were prepared for sequencing using standard NovaSeq stranded mRNA prep protocols. Sequencing was performed on the Illumina NovaSeq 6000 (GPL24676). The raw data were processed as follows: quality analysis and base quality filtering with FastQC v0.11.9 (https://www.bioinformatics.babraham.ac.uk/projects/fastqc) and Trim Galore v0.6.6 (https://github.com/FelixKrueger/TrimGalore), rRNA removing with SortMeRNA v4.3.4[51], alignment with STAR v2.7.10[52], reads counting with featureCounts[53] implemented in the Subread package v2.0.3, differential expression analysis and principal component analysis with DESeq2 v1.32[50], hierarchical clustering with hclust function in R Stats package v4.1.1[54] and functional enrichment analysis with clusterProfiler v3.10[49]. For differential expression analysis, gene raw counts were normalized with rlog function in DESeq2 v1.32[50].GOplot v1.0.2[55] was used to calculate Z-score of each enriched GO terms.

**Quantification of sample traits**. For human kidney developmental data, sample traits were collected from ArrayExpress (E-MTAB-6814). Categorical variables of samples were quantized into 1 and 0. For sex, male was quantized into 1 and female was quantized into 0. Continuous variables (Developmental stage) of samples were quantized into week according to the following rules: For embryo samples (before birth), the developmental stage value equal how many weeks they were post conception (For example, if the embryo was 10 weeks old, the value was 10.). For samples collected after birth, the developmental stage value equal 40 (human pregnancy estimated value of 40 weeks) plus age (count in weeks). 1 year was calculated as 52 weeks, 1 month as 1/12 years, and 1 week as 7 days (For example, for a sample at 6 months after birth, the development stage value was $40 + 6/12 \times 52 = 66$; for a sample at 7 days after birth, the development stage value was $40 + 7/7 = 41$) (Supplementary Data 2).

For the single-end in vitro kidney organoids differentiation RNA-seq data ($n = 41$), sample traits were collected from original sample information from GEO database (see 'In vitro kidney organoids differentiation RNA-seq data' section). For differentiation stage, samples were classified into following categories according to the description of in vitro kidney organoids differentiation[37]: cells in early differentiation stage (day 0, 4), nephron progenitor cells (day 7) and kidney organoids (days 18, 25). Categorical variables of samples (specific kidney organoids derived tissue type) were quantized into 1 and 0 (Supplementary Data 9).

**Prediction of the transcription factors**. To identify the transcription factors (TFs) that have potential synergy with CNV-lncRNA through regulating transcription, we retrieval the genomic sequence of promoter region (upstream by 2000 bases) of CNV-lncRNA from hg38 assembly with UCSC (http://genome.ucsc.edu)[56]. TFs of Homo sapiens that potentially bind to the promoter region were predicted with PROMO[57,58]. The TFs that predicted with the dissimilarity rate lower than 5% were considered as hits.

**Identification of miRNAs interactions with CNV-lncRNAs and mRNAs**. For CNV-lncRNAs distributed in the CAKUT_sig1 and CAKUT_sig2 modules (kidney developmental WGCNA, $n = 437$), we retrieved 238624 lncRNA-miRNA interaction evidences from LncBook[14]. Genomic loci (hg38) of miRNAs were annotated based on miRbase[59]. 331604 miRNA-mRNA interaction evidence were generated from our previous study[10]. Integrating the interactions evidences with human kidney developmental data ($n = 40$), we conducted the lncRNA-miRNA-mRNA regulatory network analysis using LncmiRSRN v3.0[31] to estimate the contribution of competing endogenous RNA (ceRNA) mechanism to the pathogenicity of CNV-lncRNAs distributed in the CAKUT_sig1 and CAKUT_sig2 modules.

**Statistics and reproducibility**. For RNA-seq analyzes, a threshold of adjusted $P < 0.05$ was used for differential gene expression analysis. For the correlation and causal effect in the lncRNA-miRNA-mRNA regulatory network analysis, a threshold of adjusted $P < 0.05$ was used. The $P$ values were adjusted for multiple testing using the Benjamini–Hochberg method. Hypergeometric tests were used to estimate the significance of enrichment between two gene sets, using a threshold of $P < 0.05$. For comparing expression levels of multifunctional genes between different organs and RT-qPCR analyzes, two-tailed Student's $t$ test was used for comparison between two group. For knockdown experiments on HSALNG0134318 and HSALNG0115943 in the human embryonic kidney (HEK293) cell lines, three biologically independent experiments were conducted for each group.

**Reporting summary**. Further information on research design is available in the Nature Portfolio Reporting Summary linked to this article.

## Data availability

The RNA-seq raw data for knockdown experiments on HSALNG0134318 in the human embryonic kidney (HEK293) cell lines that generated in this study are available in GEO: GSE223312. The source data for figures are available in Supplementary Data 1. Data generated during this study are available in Figshare (https://doi.org/10.6084/m9.figshare.23624658.v2, Supplementary Data 2–16 files)[60]. Accession code of other raw data that support the results in this work are all available in Methods section.

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

## Acknowledgements

We thank Ms MJ Zhou at Fresenius Medical Care for editing the English text of this manuscript. This work was supported by National Key R&D Program of China (2021YFC2701104); National Natural Science Foundation of China (82172352); National Facility for Translational Medicine (Shanghai) (TMSK-2021-133); Pudong Science Technology and Economy Commission (PKJ2022-Y04); the Collaborative Innovation Program of Shanghai Municipal Health Commission (2020CXJQ01) and Shanghai Municipal Science and Technology (20JC1418500, 20dz2260900, 20ZR1434800).

## Author contributions

Yb.L. and B.W. contributed literature review, study design, data collection, data analysis, data interpretation, and drafting the manuscript; Yy.Z., J.G., M.Q., Yw.L., and Xy.Z. contributed experimental and data interpretation; Y.X., Qh.F., and B.W. contributed supervision of all aspects of the study and manuscript preparation. All the authors have read and approved the final manuscript.

## Competing interests

The authors declare no competing interests.
