## [Peer Review File · Communications Biology]

Reviewers' comments:

Reviewer #1 (Remarks to the Author):

The article is well-structured, and scientific, with good data analysis. The CNV-lncRNA study is cutting-edge and provides some direction for future research.

1. The article focuses on two hub CNV-lncRNAs, HSALNG0134318 and HSALNG0115943. WGCNA's essential modules contain two hub CNV-lncRNAs. However, because the two hub CNV-lncRNAs are not the hub CNV-lncRNAs identified through machine learning or other methods. So, in the Introduction section, please present the current state of research on the two hub CNV-lncRNAs with a focus on renal diseases. Alternatively, please describe the method for defining hub CNV-lncRNA.

2. The brown module appears to be more relevant in the first WGCNA part; could you perhaps explain why there is no further description of the brown module?

3. Is it possible to predict CNV-lncRNA regulatory mechanisms in the study, such as miRNAs and transcription factors?

Reviewer #2 (Remarks to the Author):

In the current study, Lu and colleagues have performed a comprehensive investigation on lncRNAs located in CAKUT-associated CNVs. By integrating bioinformatic characterisation of lncRNAs in recurrent CNVs the authors have selected two lncRNA candidates HSALNG0134318 and HSALNG0115943 for further validation of the expression results. In my opinion, this is a very interesting study as the field of CAKUT research inevitably has to expand toward the non-coding part of the genome. I have minor suggestions for the improvement of the manuscript.

1) I don't find it intuitive to annotate major modules by their colours across the text (especially in the abstract).

2) CRKL has been identified as a major genetic driver of the kidney defects seen in 22q11.2 deletion syndrome. How do the authors distinct the effect of lncRNA and CRKL deletion on CAKUT development?

3) It is known that different species of ncRNA may interact or have the synergistical effect. For example, miR-185 is also located in 22q11.2 region like HSALNG0134318 and previously shown to be one of the most frequently affected miRNAs in CAKUT, besides miR-484. The authors should shortly discuss the potential synergy of the different ncRNA species located in CAKUT-associated CNVs.

4) The text needs spell-check as there are certain typos.

Reviewer #3 (Remarks to the Author):

Lu et al present an original research article entitled "Integrated analysis of copy number variation-associated lncRNAs identifies candidates contributing to the etiologies of congenital kidney anomalies". They constructed integrated analysis of copy number variation-associated lncRNAs to identify candidates contributing to the etiologies of congenital kidney anomalies (Congenital anomalies of the kidney and urinary tract (CAKUT)). (8997 candidate CAKUT associated CNV-lncRNAs and 19957 mRNAs). They used human kidney developmental transcriptome data from LncExpDB, and confirmed transcriptional regulatory effects of two CNV-lncRNAs residing in 17q12 and 22q11.2 in the kidney organoids and human embryonic kidney (HEK293) cell line. That said, the confirmatory experiments were not conducted by the authors themselves, but again relied on raw data (fastq files) of in vitro differentiation RNA-seq experiments.

All in all, the paper is well written, and the results interesting but I think one has to do some convincing biological confirmation of in silico data before publishing.

The reviewer comments are laid out below in italicized font. Our responses are given in normal blue font.

Reviewer #1 (Remarks to the Author):

The article is well-structured, and scientific, with good data analysis. The CNV-lncRNA study is cutting-edge and provides some direction for future research.

1. The article focuses on two hub CNV-lncRNAs, HSA134318 and HSA115943. WGCNA's essential modules contain two hub CNV-lncRNAs. However, because the two hub CNV-lncRNAs are not the hub CNV-lncRNAs identified through machine learning or other methods. So, in the Introduction section, please present the current state of research on the two hub CNV-lncRNAs with a focus on renal diseases. Alternatively, please describe the method for defining hub CNV-lncRNA.

R: We sincerely appreciate the valuable suggestion. 22q11.2 and 17q12 have both been discovered to be mostly associated with kidney anomalies. Based on kidney developmental transcriptomic dataset for CAKUT associated CNV-lncRNAs identification in our analyses, HSA134318 (22q11.2) and HSA115943 (17q12) were found to be the only hub lncRNAs in their corresponding CNVs regions for CAKUT modules (Supplementary data 1, 11). Therefore, we focused on the two hub CNV-lncRNAs for further validation. We have added the statement in the manuscript (**line 149-157**). In addition, we have also added detailed description for how we defined the hub CNV-lncRNAs (Module membership value ≥ 0.8) in the method part as you suggested (**line 537-539**).

#line 149-157: Although CAKUT cases share developmental origins, they encompass a broad spectrum of clinical phenotypes, such as kidney anomalies (renal agenesis, hypoplasia, dysplasia and multicystic dysplasia), vesicoureteral reflux, and obstructive uropathy, duplicated collecting system, posterior urethral valves, ectopic kidney. 22q11.2 deletion and 17q12 deletion has been most linked to kidney anomalies. Since we used the transcriptomic datasets of kidney development for coexpression analysis, we focused on 2 hub CNV-lncRNAs: HSA134318 located in 22q11.2 (MM=0.87) and HSA115943 located in 17q12 (MM = 0.85) from the CAKUT_sig1 module, which were coexpressed and significantly correlated with multiple CAKUT genes (Figure 3A, B; Supplementary Data 1, 3, 4).

#line 537-539: The module membership (MM) value was used to estimate the correlation between each gene/lncRNA and the module eigengene of a given module. The lncRNAs in each module with MM values ≥ 0.8 were defined as hub CNV-lncRNAs⁸.

2. The brown module appears to be more relevant in the first WGCNA part; could you perhaps explain why there is no further description of the brown module?

R: We thank the reviewer for pointing out this issue. In our analyses, three modules (the brown, turquoise and lightgreen modules) showed highly negative correlations with developmental stages of human kidney developmental samples (Figure 2A). The results indicated that CNV-lncRNAs in these modules tend to have high expression level in the early kidney developmental stage. Such expression pattern may imply their potential functions specific to embryonic development. Whereas,

the brown module ($P = 0.573$) did not show significant enrichment ($P < 0.05$) for known CAKUT genes (Supplementary data 1). We mainly focused on two modules that show significant enrichment for known CAKUT genes: turquoise (labeled as CAKUT_sig1, $P = 1.15 \times 10^{-6}$) and lightgreen (labeled as CAKUT_sig2, $P = 0.046$) in the present study (Figure 2C). We have added additional explanations (line 149-157) and discussed the limitation for this issue in the manuscript (line 442-448).

#line 149-157: Although CAKUT cases share developmental origins, they encompass a broad spectrum of clinical phenotypes, such as kidney anomalies (renal agenesis, hypoplasia, dysplasia and multicystic dysplasia), vesicoureteral reflux, and obstructive uropathy, duplicated collecting system, posterior urethral valves, ectopic kidney. 22q11.2 deletion and 17q12 deletion has been most linked to kidney anomalies. Since we used the transcriptomic datasets of kidney development for coexpression analysis, we focused on 2 hub CNV-lncRNAs: *HSALNG0134318* located in 22q11.2 (MM=0.87) and *HSALNG0115943* located in 17q12 (MM = 0.85) from the CAKUT_sig1 module, which were coexpressed and significantly correlated with multiple CAKUT genes (Figure 3A, B; Supplementary Data 1, 3, 4).

#line 442-448: Secondly, the significance of enrichment in known CAKUT gene sets was related to gene counts in each module and the size of known CAKUT gene sets. Since CAKUT genes that have been identified so far is limited (totaled to 172 CAKUT genes in current study, Supplementary Table 1), modules other than the CAKUT_sig1 and CAKUT_sig2 modules (e.g., the kidney developmental brown module that contained 18 CAKUT genes, $P = 0.573$; the kidney developmental magenta module that contained 8 CAKUT genes, $P = 0.076$) could not be ignored in the pathogenesis of CAKUT (Figure 2B, C; Supplementary Data 1).

3. Is it possible to predict CNV-lncRNA regulatory mechanisms in the study, such as miRNAs and transcription factors?

R: We think this is an excellent suggestion. We have performed additional analyses to identify the driver CNV-lncRNAs through the lncRNA-miRNA-mRNA competing endogenous RNA (ceRNA) regulatory mechanism (Supplementary Figure 5, 6; line 256-263). Since knockdown of *HSALNG0134318* in the HEK293 cell line affected pathways related to transcription in our experimental validation (Supplementary Figure 3B, C), we also predicted the transcription factors that might interact with *HSALNG0134318* (Supplementary Table 2; line 270-273). Discussion of synergistical effect for ncRNAs and known CAKUT genes in the 22q11.2 CNV region were also added in the manuscript (line 386-406).

#line 256-263: To reveal the potential synergy of CAKUT associated CNV-lncRNAs with other ncRNA species, we investigated the potential contribution of competing endogenous RNA (ceRNA) mechanism driven by CNV-lncRNAs distributed in the CAKUT_sig1 and CAKUT_sig2 modules (kidney developmental WGCNA, $n = 437$). We found that CAKUT gene *AFF3* in the CAKUT_sig1 module was significantly regulated by co-expressed CNV-lncRNAs *HSALNG0148020*, *HSALNG0111125* and *HSALNG0110138* through the ceRNA mechanism (Supplementary Figure 5; Supplementary Data 11). The hub CNV-lncRNA *HSALNG0134318* was

not identified as a driver in the lncRNA-miRNA-mRNA regulatory network (Supplementary Data 11).

#line 270-273: Furthermore, we predicted the synergic transcription factors (TFs) of *HSALNG0134318* according to its promoter region sequence (Supplementary Table 2; Supplementary Data 11). Among these predicted transcription factors, *TFAP2A* was in the CAKUT gene list (Supplementary Table 2; Supplementary Data 11).

#line 386-406: We identify 4 driver lncRNAs regulating CAKUT genes through ceRNA mechanism in the CAKUT_sig1 and CAKUT_sig2 modules using the LncmiRSRN²⁵. (Supplementary Figure 5). *HSALNG0134318* was not identified as a driver lncRNA since only top 20% of tested lncRNAs with enriched targeted miRNAs were selected in this algorithm. Previously, miRNA miR-185 located in 22q11.2 was shown to be associated with CAKUT²⁶. Therefore, we further investigated the potential interaction of lncRNAs and miRNAs in 22q11.2. *HSALNG0134318* is also the target of two miRNAs (has-miR-3198 and has-miR-1306-5p), of which has-miR-3198 also target the CAKUT gene *DHODH* (Supplementary Figure 6; Supplementary data 11). Whereas, knockdown of *HSALNG0134318* had no effect on the expression of *DHODH* (Supplementary Data 7). The miRNA miR-185 could targets 22q11.2 CNV-lncRNAs (*HSALNG0134144* and *HSALNG0134523*) and CAKUT genes (*ATN1*, *EMC10*, *GNB2*, *IGF1R*, *NSD1*, *SIX1* and *SIX5*) (Supplementary Figure 6; Supplementary data 11). We also noticed that three CAKUT genes including *IFG1R* ($\log_2\text{FoldChange} = -0.564$, $P_{\text{adj}} = 3.413 \times 10^{-5}$), *IFT52* ($\log_2\text{FoldChange} = 0.747$, $P_{\text{adj}} = 2.282 \times 10^{-4}$), and *CRKL* ($\log_2\text{FoldChange} = -0.569$, $P_{\text{adj}} = 1.364 \times 10^{-4}$), which could be significantly downregulated through knockdown of *HSALNG0134318*, were targets of miRNAs located at 22q11.2 (Supplementary Figure 6; Supplementary data 7). Therefore, 22q11.2 ncRNAs and mRNAs should have connections during kidney development. Deletion of such region should have synergy effect on clinical phenotypes. Since knockdown of *HSALNG0134318* affected the expression of *CRKL*, *HSALNG0134318* could be upstream of the kidney developmental pathways involving *CRKL*. Through transcription factor prediction, we identified the CAKUT gene *TFAP2A*, encoding a transcription factor, could govern the expression of *HSALNG0134318* in kidney development (Supplementary Table 2; Supplementary Data 11).

Supplementary Figure 5. CAKUT genes involved CNV-lncRNA-miRNA-mRNA regulatory network.

Supplementary Figure 6. miRNA interactions with CAKUT genes and CAKUT associated CNV-lncRNAs.

Reviewer #2 (Remarks to the Author):

In the current study, Lu and colleagues have performed a comprehensive investigation on lncRNAs located in CAKUT-associated CNVs. By integrating bioinformatic characterisation of lncRNAs in recurrent CNVs the authors have selected two lncRNA candidates HSA134318 and HSA115943 for further validation of the expression results. In my opinion, this is a very interesting study as the field of CAKUT research inevitably has to expand toward the non-coding part of the genome. I have minor suggestions for the improvement of the manuscript.

1) I don't find it intuitive to annotate major modules by their colours across the text (especially in the abstract).

R: We appreciate the valuable comments. We have re-annotated the two modules significantly enriched in CAKUT genes (the turquoise and lightgreen modules) as CAKUT_sig1 and CAKUT_sig2 (Figure 2) throughout our manuscript and the abstract (line 17-24) to make it more intuitive.

#line 17-24: correlated expression with CAKUT genes in the developing kidneys. The regulatory effects of two hub lncRNAs (HSA134318 in 22q11.2 and HSA115943 in 17q12) in the module most significantly enriched in known CAKUT genes (CAKUT_sig1) were validated experimentally. Our results indicated that the reduction of CNV-lncRNAs could downregulate CAKUT genes as predicted from our computational analyses. Furthermore, knockdown of HSA134318 would downregulate HSA115943, and affect kidney development related pathways. The results also indicated that the CAKUT_sig1 module had function significance involving multiple systems.

[Changes in Figure 2: We have re-annotated the two modules significantly enriched in CAKUT genes (the turquoise and lightgreen modules) as CAKUT_sig1 and CAKUT_sig2. We have highlighted these modules in red font]

2) *CRKL* has been identified as a major genetic driver of the kidney defects seen in 22q11.2 deletion syndrome. How do the authors distinct the effect of lncRNA and *CRKL* deletion on CAKUT development?

R: Thank you for this deep insight over our study. *CRKL* has been suggested as a genetic driver of kidney defects in 22q11.2 deletions. In fact, *CRKL* locates at the central part of 22q11.2 (22q11.21), and *HSALNG0134318* locates at the distal part of 22q11.2 (22q11.22, chr22:22,298,141-22,307,554). Since the occurrence of kidney abnormalities in 22q11.2 CNV that did not involve *CRKL* was still unexplained, the genetic effect of other candidates in 22q11.2 region should not be ignored. It's worth noting that expression of *HSALNG0134318* and *CRKL* showed highly correlated expression in developmental kidneys (Figure 3B), and knockdown of *HSALNG0134318* would significantly downregulation the expression of *CRKL* (Figure 4A). Therefore, expression disturbance of *CRKL* could be the downstream effect of *HSALNG0134318* deletion other than the dosage effect of *CRKL* itself. In general, we speculate that the effect of *HSALNG0134318* and *CRKL* on kidney development might converge to unified molecular pathways. Additional analysis and discussion over this issue has been added in the manuscript (**line 377-386**).

#line 377-386: It's worth pointing out that *CRKL* has been identified as the main genetic driver of kidney defects in 22q11.2 CNV^{23,24}, which was most frequently identified in CAKUT patients. In fact, *CRKL* locates at the central part of 22q11.2 (22q11.21), and *HSALNG0134318* locates at the distal part of 22q11.2 (22q11.22, chr22:22,298,141-22,307,554). Since the occurrence of kidney abnormalities in 22q11.2 CNV that did not involve *CRKL* was still unexplained, the genetic effect of other candidates in 22q11.2 region should not be ignored. It's worth noting that expression of *HSALNG0134318* and *CRKL* showed highly correlated expression in developmental kidneys (Figure 3B), and knockdown of *HSALNG0134318* would significantly downregulation the expression of *CRKL* (Figure 4A). Therefore, expression disturbance of *CRKL* could be the downstream effect of *HSALNG0134318* deletion other than the dosage effect of *CRKL* itself.

3) *It is known that different species of ncRNA may interact or have the synergistical effect. For example, miR-185 is also located in 22q11.2 region like HSALNG0134318 and previously shown to be one of the most frequently affected miRNAs in CAKUT, besides miR-484. The authors should shortly discuss the potential synergy of the different ncRNA species located in CAKUT-associated CNVs.*

R: Thank you for your excellent suggestion. We have performed additional analyses for potential interaction of lncRNA-miRNA and mRNA-miRNA (Supplementary Figure 5, 6). The corresponding discussion for potential synergy of different ncRNA species has been added in the manuscript (**line 386-406**).

#line 386-406: We identify 4 driver lncRNAs regulating CAKUT genes through ceRNA mechanism in the CAKUT_sig1 and CAKUT_sig2 modules using the LncmiRSRN²⁵. (Supplementary Figure

5). *HSALNG0134318* was not identified as a driver lncRNA since only top 20% of tested lncRNAs with enriched targeted miRNAs were selected in this algorithm. Previously, miRNA miR-185 located in 22q11.2 was shown to be associated with CAKUT²⁶. Therefore, we further investigated the potential interaction of lncRNAs and miRNAs in 22q11.2. *HSALNG0134318* is also the target of two miRNAs (has-miR-3198 and has-miR-1306-5p), of which has-miR-3198 also target the CAKUT gene *DHODH* (Supplementary Figure 6; Supplementary data 11). Whereas, knockdown of *HSALNG0134318* had no effect on the expression of *DHODH* (Supplementary Data 7). The miRNA miR-185 could targets 22q11.2 CNV-lncRNAs (*HSALNG0134144* and *HSALNG0134523*) and CAKUT genes (*ATN1*, *EMC10*, *GNB2*, *IGF1R*, *NSD1*, *SIX1* and *SIX5*) (Supplementary Figure 6; Supplementary data 11). We also noticed that three CAKUT genes including *IFG1R* ($\log_2\text{FoldChange} = -0.564$, $P_{\text{adj}} = 3.413 \times 10^{-5}$), *IFT52* ($\log_2\text{FoldChange} = 0.747$, $P_{\text{adj}} = 2.282 \times 10^{-4}$), and *CRKL* ($\log_2\text{FoldChange} = -0.569$, $P_{\text{adj}} = 1.364 \times 10^{-4}$), which could be significantly downregulated through knockdown of *HSALNG0134318*, were targets of miRNAs located at 22q11.2 (Supplementary Figure 6; Supplementary data 7). Therefore, 22q11.2 ncRNAs and mRNAs should have connections during kidney development. Deletion of such region should have synergy effect on clinical phenotypes. Since knockdown of *HSALNG0134318* affected the expression of *CRKL*, *HSALNG0134318* could be upstream of the kidney developmental pathways involving *CRKL*. Through transcription factor prediction, we identified the CAKUT gene *TFAP2A*, encoding a transcription factor, could govern the expression of *HSALNG0134318* in kidney development (Supplementary Table 2; Supplementary Data 11).

Supplementary Figure 5. CAKUT genes involved CNV-lncRNA-miRNA-mRNA regulatory network.

Supplementary Figure 6. miRNA interactions with CAKUT genes and CAKUT associated CNV-lncRNAs.

4) *The text needs spell-check as there are certain typos.*

R: Thank you for your advice and careful review. We have checked our resubmitted manuscript carefully and invited an English native speaker from the USA to help polish our article.

Reviewer #3 (Remarks to the Author):

Lu et al present an original rersearch article entitled "Integrated analysis of copy number variation-associated lncRNAs identifies candidates contributing to the etiologies of congenital kidney anomalies". They constructed integrated analysis of copy number variation-associated lncRNAs to identify candidates contributing to the etiologies of congenital kidney anomalies (Congenital anomalies of the kidney and urinary tract (CAKUT)). (8997 candidate CAKUT associated CNV-lncRNAs and 19957 mRNAs). They used human kidney developmental transcriptome data from LncExpDB, and confirmed transcriptional regulatory effects of two CNV-lncRNAs residing in 17q12 and 22q11.2 in the kidney organoids and human embryonic kidney (HEK293) cell line. That said, the confirmatory experiments were not conducted by the authros themselves, but again relied on raw data (fastq files) of in vitro differentiation RNA-seq experiments.

All in all, the pamer is well written, ent the results intersting but I think one has to do some convincing biological confirmation of in silico data before publishing.

R: Thanks for your suggestion. In the present study, we aimed to identify potential CNV-lncRNAs that may contribute to the etiologies of CAKUT since most current reports of CAKUT genetic factors focused on protein coding genes. Our present study was based on the assumptions that CNV may affect lncRNAs in a dosage dependent manner, thus leading to the disturbance of key co-expressed genes (known CAKUT genes) involving in normal kidney development. Therefore, we focused on broad-spectrum computational analysis in the current study and only performed RT-qPCR analyses and RNA-seq analyses in two hub CNV-lncRNAs knockdown experiments to validate that disturbance of CNV-lncRNAs could regulate the expression of CAKUT genes, as well as kidney development related pathways (Figure 4; Supplementary Figure 2, 3). We believe that further experimental investigation of the CNV-lncRNAs would help reveal novel findings for the etiologies of CAKUT associated CNV-lncRNAs We have discussed the limitation and further research proposal for this issue in the manuscript (**line 437-442**).

#line 437-442: Nevertheless, a lack of deep insight over these mechanisms is a limitation of the study. Since lncRNAs can either repress or activate gene expressions in *cis*, in *trans* or even through epigenetic modification^{6,30}, iPSC-induced kidney organoids, cardiomyocytes and neural progenitor cells in vitro models combined with high-throughput sequencing technologies such as chip-seq, scRNA-seq, and proteomics could be further used to study the molecular mechanisms by which *HSALNG0134318* and *HSALNG0115943* cause CAKUT and concomitant extrarenal malformation.

REVIEWERS' COMMENTS:

Reviewer #1 (Remarks to the Author):

I have no more questions.

Reviewer #2 (Remarks to the Author):

The authors have addressed all of the raised issues.